# Sparse Modular Activation for Efficient Sequence Modeling

**Liliang Ren**[1][*]  **Yang Liu**[2]  **Shuohang Wang**[2]  **Yichong Xu**[†]
**Chenguang Zhu**[2]  **Chengxiang Zhai**[1]
[1]University of Illinois at Urbana-Champaign    [2]Microsoft
{liliang3,czhai}@illinois.edu
{yaliu10,shuowa,chezhu}@microsoft.com
xuyc11@gmail.com

## Abstract

Recent hybrid models combining Linear State Space Models (SSMs) with self-attention mechanisms have demonstrated impressive results across a range of sequence modeling tasks. However, current approaches apply attention modules statically and uniformly to all elements in the input sequences, leading to sub-optimal quality-efficiency trade-offs. To address this limitation, we introduce *Sparse Modular Activation* (SMA), a general mechanism enabling neural networks to sparsely and dynamically activate sub-modules for sequence elements in a differentiable manner. Through allowing each element to skip non-activated sub-modules, SMA reduces computation and memory consumption of neural networks at both training and inference stages. To validate the effectiveness of SMA on sequence modeling, we design a novel neural architecture, *SeqBoat*, which employs SMA to sparsely activate a Gated Attention Unit (GAU) based on the state representations learned from an SSM. By constraining the GAU to only conduct local attention on the activated inputs, SeqBoat can achieve linear inference complexity with theoretically infinite attention span, and provide substantially better quality-efficiency trade-off than the chunking-based models. With experiments on a wide range of tasks, including long sequence modeling, speech classification and language modeling, SeqBoat brings new state-of-the-art results among hybrid models with linear complexity, and reveals the amount of attention needed for each task through the learned sparse activation patterns. Our code is publicly available at https://github.com/renll/SeqBoat.

## 1  Introduction

Recent advance on efficient sequence modeling with State Space Models (SSMs) [GGR21; GDE$^+$20; GGGR22; GB22; SWL23] has shown impressive performance for a wide range of tasks across modalities, such as text classification, image recognition and speech recognition. SSMs, as first-order linear models, defined by a set of input, output, and state variables connected by first-order differential equations, can efficiently capture the recurrent structure in sequential data with carefully designed state matrices and the application of convolutional parallelism [GGR21]. However, they still significantly underperform the self-attention [BCB14; VSP$^+$17] based model in both language modeling and machine translation [VPSP23] tasks. A recent work [FDS$^+$23] reveals that this is due to its deficiency of modeling the second-order pairwise comparisons between the input tokens, and shows that the augmentation of an additional shifted SSM layer can improve SSM's

---

[*]Work partially done during internship at Microsoft.
[†]Work done at Microsoft.

associative recalling ability. Furthermore, better quality-efficiency trade-off can be achieved by directly introducing extra self-attention modules to form a hybrid model (e.g. MEGA [MZK$^+$23] and Hybrid H3 [FDS$^+$23]) that utilizes both the first and the second order inductive biases, *i.e.*, SSM and self-attention. However, the current hybrid models apply the attention modules statically and uniformly to each of the input token regardless the property of the task itself. This can lead to sub-optimal quality-efficiency trade-offs since not all input tokens require second-order modeling and this computation need can vary substantially depending on both its context and the task difficulty.

In this paper, we aim to answer the following research questions for efficiently combining attention with SSMs:

- **RQ1:** Can neural networks learn to activate their attention modules on demand to achieve better quality-efficiency trade-off?

- **RQ2:** How much extra attention is needed for the SSMs on a task-by-task basis?

To answer these questions, we develop a new general mechanism, *Sparse Modular Activation (SMA)*, that allows a neural network to sparsely and dynamically activate its sub-modules for each of the input token in a fully differentiable manner. Specifically, we assume a neural model can be composed of multiple heterogeneous sub-modules. For the input sequence, a latent configurator sparsely maps tokens to multiple compressed sequences corresponding to sub-modules. Each sub-module is then only applied on its mapped shorter sequence. Compared with activating all sub-modules on the whole input, Sparse Modular Activation can reduce computation and memory consumption for both the training and inference stages. Notably, SMA is proved to have a full coverage of the combinatorial search space of module activation, which is further explained in Section 3.2.

Efficient learning of dynamic sparsity is notoriously difficult under the constraint of the current parallel hardware [LQC$^+$22; GZYE20; XM22]. To enable the practical efficiency gains from our module-level sparsity, we provide a simple yet efficient parallel implementation of SMA without any custom fused GPU kernels. Specifically, when compressing a batch of sequences in SMA, our implementation conducts both token selection and the sequence re-padding simultaneously using a single *scatter* operation that is widely optimized and present in modern deep learning frameworks.

To address **RQ1**, we apply SMA to construct a novel neural architecture, *SeqBoat*, that sparsely activate a Gated Attention Unit (GAU) [HDLL22] based on the state representation learned from an SSM. Both the GAU and the SSM representations are then aggregated through simple addition and activation to form a layer-level representation. Multiple same-sized SeqBoat layers are stacked sequentially to form a full neural model. Inspired by the working memory mechanism [AS68] used in human cognition, we further restrict the GAU to only apply local attention on the compressed sequence, which allows our model to have linear sequence inference complexity but theoretically infinite attention span.

We conduct comprehensive experiments to show that SeqBoat has significantly better quality-efficiency trade-off than state-of-the-art hybrid models on a wide range of tasks, including Long Range Arena (LRA) [TDA$^+$20], speech classification [War18] and language modeling [Hut06]. On the competitive LRA benchmark, SeqBoat achieves 1.96 higher average accuracy than MEGA-chunk [MZK$^+$23], the previous best hybrid model, with a 10.4$\times$ training speed up and a 95% memory reduction compared to the Transformer [VSP$^+$17] on the Text task with 4,096 input length. Thanks to the intrinsic modular sparsity brought by SMA, SeqBoat directly reveals the amount of attention needed for each data sample of each task through its sparse activation patterns of GAU, addressing **RQ2**. We demonstrate that our working memory mechanism provides substantially better computation-accuracy trade-off than chunking based models, and analyze the relationship between the working memory size and the effective attention span on various long sequence modeling tasks.

## 2  Background

To motivate and clarify our proposed techniques, we first present a mathematical formulation of our Sparse Modular Activation mechanism and show how it encompasses and generalizes previous attempts that aimed for module-level dynamic sparsity. A dedicated section for detailed comparisons between our approach with the related works is also included in Appendix F . We begin by reviewing how the standard sequence modeling is formalized to establish the common ground for our discussion.

## 2.1 Time-Invariant Sequence Modeling

Given a discrete sequence, $\mathbf{x} = \{x_1, ..., x_n\} \in \mathbb{R}^n$, consisting of $n$ tokens, a time-invariant sequence model $P_\theta$ is optimized to maximize the likelihood of the observed sequences by factorizing them as follows:

$$\max_\theta P_\theta(\mathbf{x}) = \prod_{t=1}^n P(x_t | \mathbf{x}_{<t}, \theta),$$

where $\mathbf{x}_{<t} = \{x_1, ..., x_{t-1}\}$ is the sequence history at time step $t$, and the parameter $\theta$ is independent of the time step $t$. This formulation implies that the full model parameters $\theta$ and the full history $\mathbf{x}_{<t}$ are both essential for the conditional prediction of each token $x_t$. However, one potential issue is as the prediction difficulty of each token may differ depending on the context and the position, this static model $P_\theta$ can lead to sub-optimal accuracy-efficiency trade-off by wasting computation on either unimportant context [SJP$^+$21] or easy-to-predict tokens [Gra16].

# 3 Learning Sparse Modular Activation

To cover a larger search space that may contain more efficient sequence models, we propose to formulate sequence modeling as a problem of finding an optimal time-variant model that can dynamically activate a subset of modules from a pre-defined function space for each time step.

## 3.1 Time-Variant Sequence Modeling

Formally, a time-variant sequence model is defined on a compact function space $\mathcal{F} : \mathcal{X}_t^c \mapsto [0,1]^{n \times V}$, where $V$ is the size of the vocabulary and $\mathcal{X}_t^c = \{\mathbf{x}_t^c : \mathbf{x}_t^c \subseteq \mathbf{x}_{<t} \in \mathcal{X} \subseteq \mathbb{R}^n\}$, contains all possible sub-sequences of the sequence history $\mathbf{x}_{<t}$. Then for each of the token prediction at the time step $t$, the model learns to apply a function $f_t \in \mathcal{F}$ with the parameters $\theta_t$ that maximizes the sequence probability, *i.e.*,

$$\max_{f_t, \theta_t, \mathbf{x}_t^c} P_\mathcal{F}(\mathbf{x}) = \prod_{t=1}^n P_{f_t}(x_t | \mathbf{x}_t^c, \theta_t) \quad \text{s.t.} \quad \mathbf{x}_t^c \subseteq \mathbf{x}_{<t} \tag{1}$$

This formulation generalizes the previous works in pursuing a dynamic and sparse model for sequence modeling, where the connections are further explained in Appendix F. In this work, we assume the function space $\mathcal{F}$ is chain-structured, *i.e.*, $\mathcal{F} = \mathcal{H} \circ \mathcal{L}_N \circ \cdots \circ \mathcal{L}_1 \circ \mathcal{E}$, where $\mathcal{H} : \mathbb{R}^{n \times d_m} \mapsto [0,1]^{n \times V}$ is the classification function, $\mathcal{E} : \mathbb{R}^n \mapsto \mathbb{R}^{n \times d_m}$ is the embedding function, $N$ is the number of intermediate layers, $d_m$ is the model size and $\mathcal{L} : \mathbb{R}^{n \times d_m} \mapsto \mathbb{R}^{n \times d_m}$ is the function space of the intermediate mappings. We further assume that $\mathcal{L}$ is the spanning set of a finite number of the function $f_i^l$ with its parameters $\theta_i^l$, *i.e.*, $\mathcal{L} = \text{span}\{f_1^l, ..., f_M^l\}$, where $M$ is the number of pre-defined functions. These assumptions justify the design of our Sparse Modular Activation mechanism, which is further explained in the following section.

## 3.2 Sparse Modular Activation

Sparse Modular Activation (SMA) introduces a latent configurator at each time step $t$ and each layer of a neural sequence model. The configurator produces a binary decision vector $\mathbf{a}_t \in \{0,1\}^M$ to only activate those functions $f_i^l$ that have the decision value $a_t^i = 1$. Only the activated function will be applied to process the input representations and return the outputs. These outputs are then aggregated with a linear combination to generate the final vector, $\mathbf{y}_t$, of the layer. To enable end-to-end differentiability, we also require the latent configurator to output the confidence probability of its decisions, $\mathbf{c}_t \in [0,1]^M$, and respectively scale the output from the activated function with its confidence values $c_t^i$. The function space of a layer equipped with SMA can be described as follow,

$$\mathcal{L}_{\text{SMA}}(\mathbf{a}_t, \mathbf{c}_t) = \left\{ \sum_{i \in I} \zeta_i c_t^i f_i^l \mid \forall \zeta_i \in \mathbb{R}, I = \{i \mid a_t^i = 1, 1 \le i \le M\} \right\}$$

where $I$ is the index set of activated functions. Notably, SMA has the following important property,

**Theorem 1** (Function Space Coverage of SMA). *For any $\mathcal{L}' \subseteq \mathcal{L} = span\{f_1^l, ..., f_M^l\}$, there exists a pair of $(\mathbf{a}_t', \mathbf{c}_t')$ that $\mathcal{L}_{SMA}(\mathbf{a}_t', \mathbf{c}_t') = \mathcal{L}'$. In other words, SMA has a **full** coverage of the function space $\mathcal{L}$.*

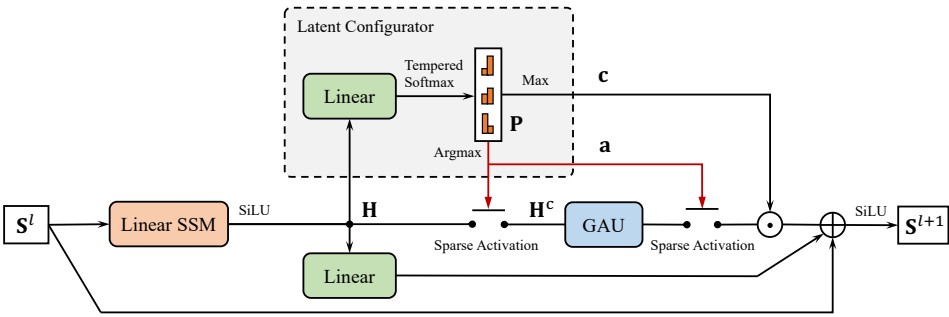

Figure 1: The proposed SeqBoat layer. The black lines indicate that the gradients can be back-propagated and the red lines stand for gradients blocking. $\odot$ means the element-wise multiplication, and $\oplus$ is the point-wise addition. The *max*, *argmax* and *softmax* operators are all applied to the projected dimension after the linear layer in the latent configurator block. The sparse activation operators are respectively instantiated as *compress* and *extract* operators for parallel processing.

The proof is provided in Appendix B. Intuitively, we can see that the original function space $\mathcal{L}$ is recovered, *i.e.*, $\mathcal{L}_{\text{SMA}} = \mathcal{L}$, if all the functions are activated, and the model can adaptively select any subspace of $\mathcal{L}$ through controlling the binary decision vector $\mathbf{a}_t$. During sequential inference stage, for each function that requires the contextual information, we keep a record of its activated input representations across the time steps to form a memory $\mathbf{H}_t^c \subseteq \mathbf{H}_{<t} \in \mathbb{R}^{n \times d_m}$ (where $\mathbf{H}_{<t}$ is the input sequence history), so that each function can recall its contextual memory for each of the decoding step. During parallel training stage, the activated input representations for each function (or module) are sparsely selected to compose the compressed sequence $\mathbf{H}^c$ for the module to conduct parallel processing, and this process is implemented as a *compress* operator. The returned representations are then mapped to its original input position in a new sequence whose inactivated positions are filled with zero vectors, which is denoted as an *extract* operator. We illustrate this parallel implementation in Figure 2 for better understanding.

The Pytorch-like [PGM+19] code snippets of the *compress* and *extract* operators are provided in the Appendix A.1 with an efficient support of batched sequences using the *scatter* operation.

In this work, we explore applying SMA to address our **RQs**, in which the following two types of functions in the function space $\mathcal{L}$ are considered: (1) a linear State Space Model (SSM) [GGGR22; MZK+23] that efficiently captures the recurrent sequential structure, and (2) a Gated Attention Unit (GAU) [HDLL22] that performs second-order pairwise comparisons between the sequence elements. We also keep the SSM as an always-activated module and only apply SMA to the GAU, and in this case the effective number of modules $M = 1$. This is because for long sequence modeling, the main computation and memory bottleneck comes from the second-order self-attention and the gating operations inside the GAU module. We modularize the combination of SSM, GAU and SMA as a neural layer, which we call a SeqBoat layer. Its details are further explained in the next section.

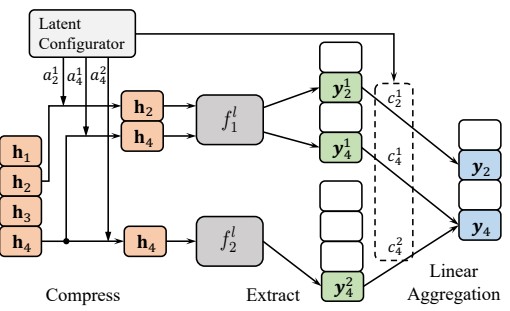

Figure 2: The proposed parallel implementation of the Sparse Modular Activation (SMA) mechanism. In this example, we have two modules $f_1^l, f_2^l$ and an input sequence $\mathbf{H} = \{\mathbf{h}_1, \ldots, \mathbf{h}_4\}$. We assume only $a_2^1, a_4^1, a_4^2$ have values equal to one. The white block means a zero vector. The compressed sequences for modules $f_1^l$ and $f_2^l$ are $\mathbf{H}_1^c = \{\mathbf{h}_2, \mathbf{h}_4\}$ and $\mathbf{H}_2^c = \{\mathbf{h}_4\}$ respectively. The final outputs are aggregated as $\mathbf{y}_4 = c_4^1 \mathbf{y}_4^1 + c_4^2 \mathbf{y}_4^2$, and $\mathbf{y}_2 = c_2^1 \mathbf{y}_2^1$.

## 3.3 Model Architecture

Figure 1 illustrates the proposed architecture for our SeqBoat layer. Given an input sequence representation $\mathbf{S}^l \in \mathbb{R}^{n \times d_m}$ for the $l$-th layer, where $n$ is the length of the sequence and $d_m$ is the hidden size, we first apply a linear time-invariant State Space Model (SSM) [GGR21] over the sequence

followed by a SiLU activation [EUD17] to obtain the state representations $\mathbf{H} \in \mathbb{R}^{n \times d_m}$,

$$\text{SSM}(\mathbf{S}^l) = \mathbf{K} * \mathbf{S}^l + \mathbf{D} \odot \mathbf{S}^l \tag{2}$$
$$\mathbf{H} = \text{SiLU}(\text{SSM}(\mathbf{S}^l))$$

where $*$ is a convolution operator which is efficiently implemented using FFTs [GGR21], $\odot$ indicates the element-wise multiplication, $\mathbf{D} \in \mathbb{R}^{1 \times d_m}$ is the learnable parameters, and $\mathbf{K} \in \mathbb{R}^{n \times d_m}$ is the SSM convolution kernel. In this work, we use the Multi-dimensional Damped Exponential Moving Average (MD-EMA) proposed by MEGA [MZK$^+$23] for the kernel parameterization, whose details are included in Appendix A.3. Note that our work is orthogonal to the SSM kernel design as long as it can provide an efficient computation of the recurrent state representation. It is possible that using a more advanced SSM such as Liquid-S4 [HLW$^+$22] or S5 [SWL23] can lead to better performance, but we keep using MD-EMA for a fair comparison with MEGA. Given the state representation $\mathbf{H} \in \mathbb{R}^{n \times d_m}$, a latent configurator with one linear layer is applied to obtain the decision vector $\mathbf{a} \in \{0, 1\}^n$ and the confidence probability vector $\mathbf{c} \in [0, 1]^n$ with the Tempered Softmax, *i.e.*,

$$\mathbf{p}_i = \frac{e^{(\mathbf{H}\mathbf{w}_i + b_i)/\tau}}{e^{(\mathbf{H}\mathbf{w}_0 + b_0)/\tau} + e^{(\mathbf{H}\mathbf{w}_1 + b_1)/\tau}} \quad \text{for } i \in \{0, 1\}$$
$$\mathbf{a} = \arg\max_i \mathbf{p}_i, \quad \mathbf{c} = \max_i \mathbf{p}_i$$

where $\mathbf{w}_0, \mathbf{w}_1 \in \mathbb{R}^{d_m}, b_0, b_1 \in \mathbb{R}$ are learnable parameters, $\tau \in \mathbb{R}_{>0}$ is a learnable temperature parameter which is initialized as $\alpha\sqrt{d_m}$ (where $\alpha$ is a hyperparameter denoted as the initial temperature scale). Previous works [RSVG20a; GLL$^+$22; FZS22; RZW$^+$22] often seek applying auxiliary losses to regularize the exploration and the exploitation of the latent decisions. However, in this work, we don't apply any auxiliary losses and empirically observe that solely using Tempered Softmax works well for SMA. To provide an explanation of this phenomenon, we show that the implicit regularization [BD20] of gradient decent will conditionally encourage the exploration of the latent decisions if the Softmax function with learnable temperature (i.e. Tempered Softmax) is used for decision confidence calculation. The details of the theorem and the proof are provided in Appendix C.

Based on the decision values $\mathbf{a}$, a Gated Attention Unit (GAU) is activated and applied on the compressed sequence $\mathbf{H}^c$ resulted from the compress operator as previously described in Section 3.2,

$$\mathbf{H}^c = \text{Compress}(\mathbf{H}, \mathbf{a}) \qquad \in \mathbb{R}^{r \times d_m}$$
$$\mathbf{Y}^c = \text{GAU}(\mathbf{H}^c) \qquad \in \mathbb{R}^{r \times d_m}$$
$$\mathbf{Y} = \text{Extract}(\mathbf{Y}^c, \mathbf{a}) \qquad \in \mathbb{R}^{n \times d_m}$$

where $r = \sum_{t=1}^n a_t \leq n$ is the total number of the time steps when the GAU module is activated. We use Squared ReLU [SML$^+$21] for the attention function on the image tasks and Softmax for the text related tasks. The detailed implementation of GAU is shown in Appendix A.2. With the extracted output $\mathbf{Y} \in \mathbb{R}^{n \times d_m}$, the final layer output is computed with a linear aggregation followed by a non-linear activation, *i.e.*,

$$\mathbf{S}^{l+1} = \text{SiLU}(\mathbf{c} \odot \mathbf{Y} + \mathbf{H}\mathbf{W} + \mathbf{b} + \mathbf{S}^l)$$

where $\mathbf{W} \in \mathbb{R}^{d_m \times d_m}, \mathbf{b} \in \mathbb{R}^{d_m}$ are the learnable parameters. We also add a normalization layer either before the Linear SSM (*i.e.*, Pre-Norm) in the residual branch or after the final layer output $\mathbf{S}^{l+1}$ (*i.e.*, Post-Norm) to stabilize the training process. The SeqBoat model is then constructed as stacking $N$ number of SeqBoat layers of the same hidden size $d_m$ with the task specific embedding and classification layers. Notably, our model does **not** have any Feed-Forward Networks (FFN) anywhere in the architecture.

**Working Memory Mechanism** The computation complexity of our model is dynamically dependent on the activation probability of the GAU module $p = r/n = \sum_{t=1}^n a_t/n$ for each of the SeqBoat layer. This means that the computation complexity is $O(r^2 + n\log(n))$ for parallel training and $O(r^2 + n)$ for auto-regressive decoding. While it is possible that the number of the GAU-activated time steps is far smaller than the length of the full sequence for a trained model, a random initialized SeqBoat model will still activate the GAU module for $n/2$ number of times on average, leading to a non-ideal $O(n^2)$ training complexity at the early stage of training. To solve this problem, we restrict the GAU to only conduct local attention [AOA$^+$20] with a sliding window of size

$w \ll n$ on the compressed sequence $\mathbf{H}^c$ during the parallel training stage. Each token will attend to the nearest $w$ tokens in the compressed sequence, *i.e.*, $w/2$ past tokens and $w/2$ future tokens for bidierectional sequence modeling, and $w$ number of past tokens for auto-regressive sequence modeling. This approach keeps a First-In-First-Out (FIFO) memory with size $w$ of the compressed sequence history during auto-regressive decoding, which can be understand as a working memory [AS68] for the GAU module. With this mechanism, SeqBoat reduces the training complexity to $O(rw + n \log(n))$ and the inference complexity to $O(rw + n)$, while maintaining the ability of attending to the key tokens whose distances from the query are longer than the working memory size $w$. This is because the elapsed time steps between two GAU activations is generally not bounded by the size of the working memory.

## 4 Experiments and Results

We evaluate the SeqBoat model on three representative long-range sequence modeling benchmarks across image, text and speech modalities to have a comprehensive understanding of its capacity. The implementation details of our model for each of the tasks are presented in Appendix A.

Besides model performance on task metrics, we also report training speed and memory consumption as the efficiency of each model. Since the dynamic sparsity of SeqBoat will affect both the speed and the memory consumption throughout the training process, we measure its training speed up as the average per step wall time across the full training stage, and the memory consumption as the average per step GPU memory allocation. More details of the efficiency measurement are included in Appendix A.6.

### 4.1 Baseline Models

We compare our proposed SeqBoat model with multiple strong baseline models besides the Transformer [VSP+17] and its variant Transformer-XL [DYY+19].

**S4** [GGR21] is a linear State Space Model (SSM) that efficiently captures the recurrent structure in sequential data. It employs a new parameterization for the state space model, enabling efficient and principled modeling of long-range dependencies. S4D-LegS [GGGR22] is a simpler variant of S4 that uses a diagonal state matrix.

**MEGA** [MZK+23] is a hybrid model that combines a gated self-attention mechanism with a Multi-dimensional Damped Exponential Moving Average (MD-EMA) parameterized SSM. It aims to achieve a better quality-efficiency trade-off by utilizing both the first and second-order inductive biases from SSM and self-attention. MD-EMA can be considered as a simpler variant of S4 that is similar to S4D. MEGA also has a linear complexity variant, MEGA-chunk, that applies attention to each local chunk of fixed length.

**H3** [FDS+23] introduces a hybrid SSM layer that stacks a diagonal SSM and a shifted SSM to model the comparisons between the elements in a sequence. It also provides a hybrid model that includes extra self-attention layers for better language modeling performance.

**S5** [SWL23] is the state-of-the-art SSM model that improves over S4 with multi-input multi-output and time-domain processing. It leverages the parallel scan operator in JAX [BFH+18] for an efficient parrallel implementation of recurrence unrolling.

**Liquid-S4** [SWL23] extends over S4 through allowing the state transition to be input-dependent. The proposed approach can be cast back to the convolutional framework and re-use the Cauchy kernel for efficient computation.

**Reformer** [KKL20] is a memory-efficient variant of the Transformer that uses locality-sensitive hashing (LSH) to perform approximate nearest neighbor search for attention computation.

**Performer and Linformer** are two efficient variants of the Transformer model. The Performer [CLD+20] uses the Fast Attention via positive Orthogonal Random features algorithm to approximate the self-attention mechanism, reducing the computational complexity. The Linformer [WLK+20] employs low-rank approximations to project the self-attention mechanism into a lower-dimensional space, resulting in a more efficient system.

Table 1: Test accuracy on the LRA benchmark. We report the relative training speed up and memory reductions on the Text task with 4,096 sequence length. * indicates a hybrid model. "Retr." and "Path." are the abbreviations of Retrieval and the Pathfinder tasks respectively. Best results of linear inference complexity models are in bold, second best are underlined.

| Models | ListOps | Text | Retr. | Image | Path. | Path-X | Avg. | Speed | Mem. |
|---|---|---|---|---|---|---|---|---|---|
| *Quadratic Inference Complexity* | | | | | | | | | |
| Transformer | 37.11 | 65.21 | 79.14 | 42.94 | 71.83 | ✗ | 59.24 | 1× | 1× |
| MEGA* | 63.14 | 90.43 | 91.25 | 90.44 | 96.01 | 97.98 | 88.21 | 2.9× | 0.31× |
| *Sub-quadratic Inference Complexity* | | | | | | | | | |
| Reformer | 37.27 | 56.10 | 53.40 | 38.07 | 68.50 | ✗ | 50.67 | 0.8× | 0.24× |
| BigBird | 36.05 | 64.02 | 59.29 | 40.83 | 74.87 | ✗ | 55.01 | 1.1× | 0.30× |
| **SeqBoat-full**\* | 61.65 | 89.60 | 91.67 | 89.96 | 95.87 | 95.28 | 87.33 | 6.2× | 0.07× |
| *Linear Inference Complexity* | | | | | | | | | |
| Performer | 18.01 | 65.40 | 53.82 | 42.77 | 77.05 | ✗ | 51.41 | 5.7× | 0.11× |
| Linformer | 35.70 | 53.94 | 52.27 | 38.56 | 76.34 | ✗ | 51.36 | 5.5× | 0.10× |
| Luna-256 | 37.98 | 65.78 | 79.56 | 47.86 | 78.55 | ✗ | 61.95 | 4.9× | 0.16× |
| S4 | 59.10 | 86.53 | 90.94 | 88.48 | 94.01 | 96.07 | 85.86 | 4.8× | 0.14× |
| S4D-LegS | 60.47 | 86.18 | 89.46 | 88.19 | 93.06 | 91.95 | 84.89 | 6.1× | 0.14× |
| S5 | 62.15 | 89.31 | **91.40** | 88.00 | 95.33 | **98.58** | 87.46 | 6.1× | 0.14× |
| Liquid-S4 | **62.75** | 89.02 | 91.20 | 89.50 | 94.8 | 96.66 | 87.32 | 1.2× | 0.17× |
| H3* | 57.50 | 88.20 | 91.00 | 87.30 | 93.00 | 91.80 | 84.80 | 6.0× | 0.24× |
| MEGA-chunk* | 58.76 | **90.19** | 90.97 | 85.80 | 94.41 | 93.81 | 85.66 | 7.0× | 0.09× |
| **SeqBoat**\* | 61.70 | 89.60 | 91.28 | **90.10** | **96.35** | 96.68 | **87.62** | 10.4× | 0.05× |

**Luna** [MKW$^+$21] is a linear unified nested attention mechanism that approximates Softmax attention with two nested linear attention functions, resulting in linear time and space complexity.

## 4.2 Long Sequence Modeling

We first evaluate SeqBoat and SeqBoat-full, a variant of SeqBoat conducting the full attention over the compressed sequence without the working memory mechanism, on the Long Range Arena (LRA) benchmark [TDA$^+$20]. LRA is designed to test the ability of neural models to capture long-range dependencies in various modalities with the following six tasks: ListOps [NB18], text classification (Text; [MDP$^+$11]), document retrieval (Retrieval; [RMQAJ13]), image classification (Image; [KH$^+$09]), Pathfinder [LKV$^+$18] and its longer-range variant Path-X.

From Table 1, we can see that compared with other hybrid models with linear inference complexity, SeqBoat achieves the best average accuracy across all tasks, with a 10.4× speed up and a 95% memory reduction compared to the Transformer. When compared with a pure SSM-based model, S4, which has more general modeling power than the MD-EMA used in our model, SeqBoat achieves a substantially higher accuracy on average with a 2.17× speed up and a 64% memory reduction. Our model even achieves a significantly better average accuracy than state-of-the-art SSM model, S5 [SWL23], with a 0.16 absolute improvement, resulting in a new high score for models with linear inference complexity. Surprisingly, SeqBoat outperforms SeqBoat-full on Pathfinder and the Path-X tasks. This demonstrates the effectiveness of our proposed working memory mechanism to efficiently capture long-range dependencies, which is further analysed in Section 5. Since the learned module-level sparsity of our model is dynamic and task-specific, we also include the measurements for training and inference speedup and training memory allocation for all the tasks of LRA in Appendix D to provide a holistic view of the efficiency of our model.

## 4.3 Speech Classification

We also evaluate SeqBoat on the long-range speech classification task using the Speech Commands dataset [War18]. We follow S4 [GGR21] to use the SC10 subset of the dataset. It is a 10-class

classification task of raw speech which contains sequences of length 16k. As shown in Table 2, SeqBoat achieves competitive performance compared to the state-of-the-art S4 model and MEGA-chunk, with a 1.32× speed up and a 56% memory reduction compared to MEGA-chunk.

Table 2: Test accuracy on the Speech Commands 10-way classification task of raw audio (Input length 16,000). We also report the training speed up and the memory consumption reduction compared with MEGA-chunk.

| Model | #Param. | Acc. (↑) | Speed (↑) | Mem. (↓) |
|---|---|---|---|---|
| S4 | 300K | **97.50** | - | - |
| MEGA-chunk | 300K | 96.92 | 1.00× | 1.00× |
| **SeqBoat** | 293K | 97.35 | 1.32× | 0.44× |

### 4.4 Language Modeling

We evaluate our model on enwik8 [Hut06], a popular language modeling benchmark. It consists of the first 100 million bytes of the English Wikipedia and is commonly used to evaluate the performance of sequence models on long-range dependencies. We follow previous studies to use it as a character-level language modeling task with a vocabulary size of around 200. During inference, we evaluate our model on the test set of enwik8 with a sequence length of 8,192 tokens and each token is equipped with a minimum of 7,000 tokens context window. We report the bits per character (BPC) metric for evaluation. As shown in Table 3, we compare SeqBoat with Transformer-XL [DYY+19], Adaptive Span [SGBJ19], and MEGA-chunk. SeqBoat achieves the same performance as the previous best hybrid model, MEGA-chunk, but with a significant 1.16× training speed up. This demonstrates the effectiveness of our proposed model in auto-regressive language modeling of long texts.

Table 3: Model performance on the enwik8 character-level language modeling task with a 8192 training sequence length. Training speed up and memory consumption reduction compared with MEGA-chunk are also reported.

| Model | #Param. | bpc (↓) | Speed (↑) | Mem. (↓) |
|---|---|---|---|---|
| Transformer-XL | 41M | 1.06 | - | - |
| Adaptive Span | 39M | 1.02 | - | - |
| MEGA-chunk | 39M | 1.02 | 1.00× | 1.00× |
| **SeqBoat** | 39M | 1.02 | 1.16× | 1.07× |

## 5 Analysis

In this section, we analyze experimental results of SMA from several research aspects. We also include comprehensive ablation studies of our model in Appendix E.

**How much attention is needed for different sequence modeling tasks?** As shown in Figure 3, we draw the activation time of the GAU module at different layers of our SeqBoat-full models for each of the task in the LRA benchmark. We measure the mean and the standard deviation (plotted as error bars) of the activation time on 100 sequences randomly sampled from the validation set of each task. The lines seem to be truncated due to the fact that different models have different number of layers. Generally, we can see that Image (in Blue and Circle) and Pathfinder (in Green and Triangle) need significantly more attentions than the text-

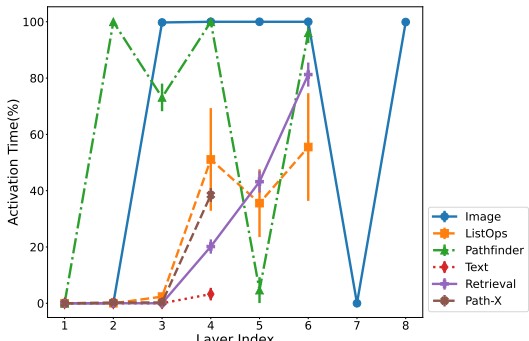

Figure 3: The activation time (with error bars) of the GAU module at different layers of the SeqBoat-full model for different tasks in the LRA benchmark.

based tasks (*i.e.*, Text, Retrieval and ListOps). We can also see that our model trained on ListOps (in Orange and Square) has a much larger variance of GAU activation time than other tasks, which

may reflect that the task difficulty per data sample of ListOps has a larger variance than other tasks. Another trend we can see is that the bottom layers near the inputs need much less attention (and sometimes even drop the attention module entirely) than the high level layers. We conjecture that this is because the SSMs with first-order complexity is already capable to capture the low-level features while the attention modules are more demanded at higher levels for conducting more complex computations of second-order pairwise comparisons.

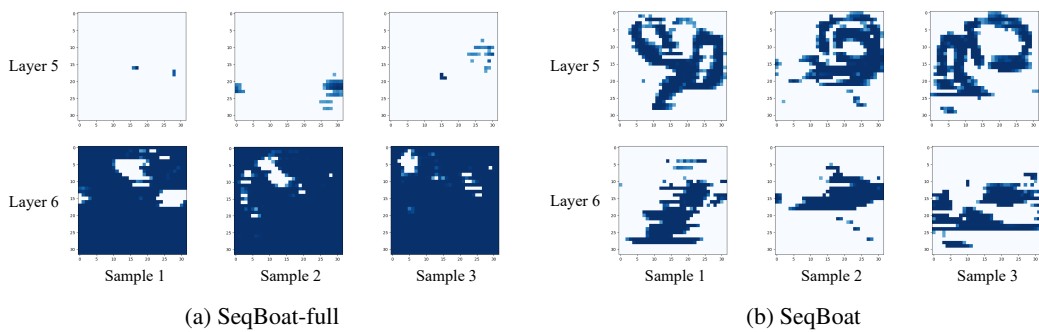

Figure 4: The confidence probabilities of the GAU modular activation at each time step for the last two layers of the SeqBoat-full and the SeqBoat model. The results are measured on three input sequences randomly sampled from the validation set of the Pathfinder task. The sequences are reshaped back to $32 \times 32$ squares for better visualization. Darker the blue color, higher the confidence. The white blocks indicate the time steps when GAUs are not activated.

**How is the learned sparse modular activation distributed over the input sequence?** As shown in Figure 4, we draw the confidence probabilities of the sparse modular activation for both the SeqBoat-full and the SeqBoat models on the Pathfinder task from the LRA benchmark. Generally, we can see that different input sequences has different modular activation patterns, which indicates that our modular activation is dynamically sparse. We can also see that the SeqBoat activation pattern is learned to be more structured than the SeqBoat-full, which may explain the superior performance of SeqBoat over SeqBoat-full on the Pathfinder task. Another trend we can see is that different layers have their own patterns: *e.g.*, the Layer 5 of the SeqBoat model seems trying to find the curly paths in the latent representation space, while the Layer 6 seems aggregating the results from the contiguous time steps.

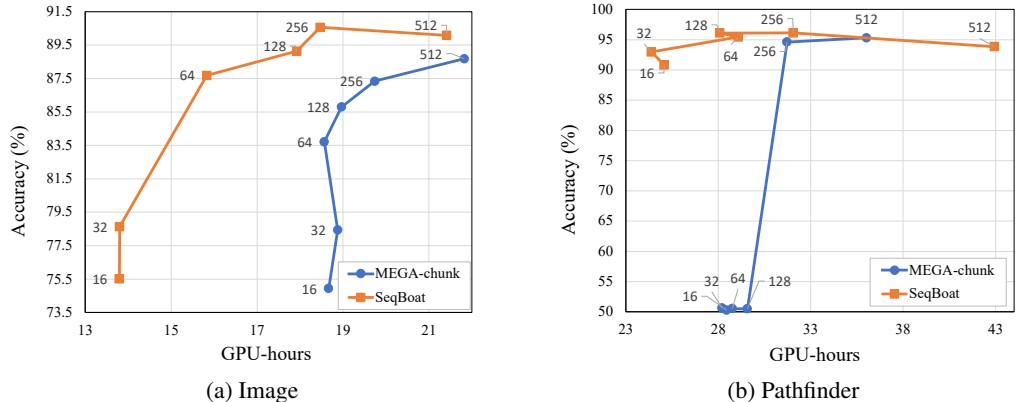

Figure 5: Training Speed v.s. Validation Accuracy trade-off on Image and Pathfinder of the LRA benchmark for different models with varying memory/chunk sizes. SeqBoat keeps a working memory of the compressed sequence, while MEGA-chunk splits the input sequence into non-overlapping sub-sequences. The memory/chunk sizes are marked along the lines. The GPU-hours for Image are measured on NVIDIA RTX A5000 GPUs, and Pathfinder on V100 GPUs with 32GB memory.

**How does the working memory size affect the speed-quality trade-off?** We compare the trade-off ability of our SeqBoat with MEGA-chunk on Image and Pathfinder tasks of the LRA benchmark,

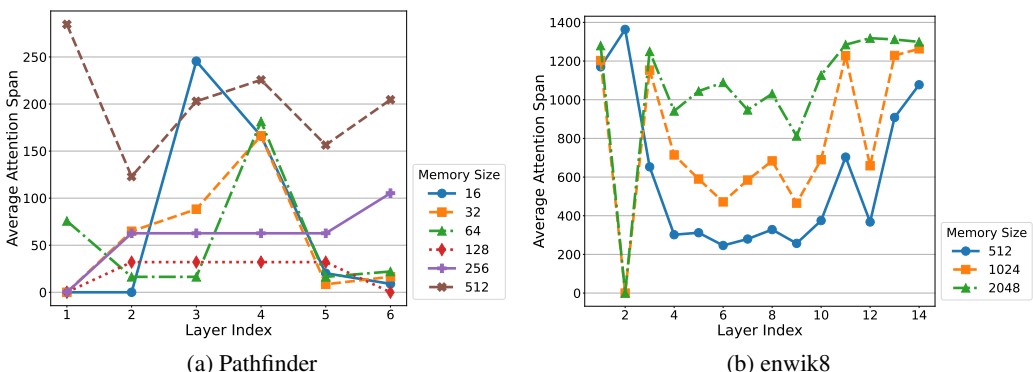

(a) Pathfinder  (b) enwik8

Figure 6: Average Attention Span v.s. Layer Index on both the Pathfinder and the enwik8 tasks for different SeqBoat models with different working memory sizes. A smaller layer index indicates the layer is closer to the input. For reference, the average attention span of a sliding window based local attention is the half of the window size.

as shown in Figure 5. We draw the Pareto frontiers by respectively varying the working memory size for SeqBoat and the chunk size for MEGA-chunk from the set $\{16, 32, 64, 128, 256, 512\}$. The results are averaged with three random seeds. We can see that our models have significantly better trade-off than MEGA-chunk models for both the Image and the Pathfinder tasks. Notably, the Mega-chunk models generally fail (achieving around 50% accuracy as random guessing) on Pathfinder tasks with less than 128 chunk size, while our SeqBoat can still achieve around 90% accuracy with only a memory size of 16. This indicates the effectiveness of our working memory mechanism in capturing long term interactions.

**How does the size of working memory affect the effective attention length of the GAU module?**
We draw Figure 6 to investigate the relationship between the working memory size and the average effective attention length on both Pathfinder and the enwik8 datasets. We first measure the average attention distance from the query token to its key tokens for each of the time step, and the results are further averaged for all the time steps in a sequence. The average attention span on Pathfinder is measured on 100 sequences randomly sampled from the validation set. The average attention span on enwik8 is calculated based on a sequence of length 8,192 sampled from the validation set, and each token is equipped with a minimum of 7,000 tokens context window. From the figures, we can see a general trend that the models with a small working memory size (Size 16, 32 and 64 for Pathfinder, and Size 512 for enwik8) can have a far longer average attention span than its memory size. Notabaly, the layer 3 of the SeqBoat model with only 16 working memory slots surprisingly achieves an average attention span of 245 for the Pathfinder task. This partially explains the superior performance of our model over MEGA-chunk with small memory sizes and directly proves that our working memory mechanism can efficiently enable long-term interactions between the sequence elements.

## 6 Conclusion

In this paper, we present Sparse Modular Activation (SMA), a general mechanism for sparsely and dynamically activating sub-modules in neural networks. SMA not only has an efficient parallel implementation but also provides a theoretical guarantee for a full coverage of the function space of multiple modules. To address the challenges of combining attention with Linear State Space Models (SSMs), we apply SMA to develop a novel neural architecture, SeqBoat, that sparsely activates a Gated Attention Unit (GAU) based on the state representations from an SSM. The proposed model demonstrates a significantly better quality-efficiency trade-off compared to existing hybrid models across various sequence modeling tasks. To the best of our knowledge, SMA is the first mechanism that allows a neural network to obtain practical efficiency and complexity gains from sparsely activating a self-attention-like module. The strong empirical performance show that even with a preliminary application of SMA, we could already see substantial efficiency and interpretability benefits. For future work, we mainly want to explore the scaling behavior of SMA and also how to incorporate SMA with the pre-trained large language models.

## Acknowledgement

We would like to thank Canwen Xu, Ziyi Yang, Ruochen Xu and Azure Cognitive Services Research group members for their feedback. We also want to thank Qi Zeng and Zixuan Zhang for the early discussion of the project.

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

# A  Implementation Details

## A.1  Code for Compress and Extract Operators in SMA

```python
def compress(q, a):
    # Args:
    #   q: Input sequences with the shape:
    #       (Batch Size, Input Sequence Length, Hidden Size)
    #   a: Binary decision values with the shape:
    #       (Batch Size, Input Sequence Length, 1)
    # Returns:
    #   Compressed sequences with the shape:
    #       (Batch Size, Compressed Sequence Length, Hidden Size)

    # Max sequence length of the compressed sequences
    max_sl = a.sum(-2).max()
    # Calcuate the indices in the compressed sequence
    index_q = a * torch.cumsum(a,dim=-2)

    # Simultaneously map and pad to build the compressed sequences
    # +1 for temporarily storing non-activated elements
    shape = q.shape[:-2] + (max_sl+1,) + q.shape[-1:]
    new_q = torch.zeros(shape)
    new_q.scatter_(-2,index_q, q)
    new_q = new_q[:,1:,:] # Non-activated elements are discarded

    return new_q
```

Listing 1: Pytorch-like code snippet for the Compress operator.

```python
def extract(h, a):
    # Args:
    #   h: Compressed sequences with the shape:
    #       (Batch Size, Compressed Sequence Length, Hidden Size)
    #   a: Binary decision values with the shape:
    #       (Batch Size, Input Sequence Length, 1)
    # Returns:
    #   Output sequences with zeros for non-activated positions:
    #       (Batch Size, Input Sequence Length, Hidden Size)

    # Calcuate the indices in the compressed sequence
    index_q = a * torch.cumsum(a,dim=-2)
    # Pad the sequence dimension to account for non-activated positions
    h = F.pad(h, (0,0,1,0))
    h = torch.gather(h,-2,index_q.expand(-1,-1,h.shape[-1]))
    return h
```

Listing 2: Pytorch-like code snippet for the Extract operator.

## A.2  Details of Gated Attention Unit

Our Gated Attention Unit (GAU) mostly follows the original implementation as [HDLL22], with slight modifications of the relative position bias. Given the compressed sequence $\mathbf{H}^c \in \mathbb{R}^{r \times d_m}$, we first compute the query, key and value sequences, $\mathbf{Q}, \mathbf{K}, \mathbf{V}$, *i.e.*,

$$\mathbf{Q} = \mathbf{w}_q \odot \mathbf{H}^c + \mathbf{b}_q \qquad \in \mathbb{R}^{r \times d_z} \qquad (3)$$

$$\mathbf{K} = \mathbf{w}_k \odot \mathbf{H}^c + \mathbf{b}_k \qquad \in \mathbb{R}^{r \times d_z} \qquad (4)$$

$$\mathbf{V} = \text{SiLU}(\mathbf{H}^c \mathbf{W}_v + \mathbf{b}_v) \qquad \in \mathbb{R}^{r \times d_v} \qquad (5)$$

where $\mathbf{w}_q, \mathbf{w}_q, \mathbf{b}_q, \mathbf{b}_k \in \mathbb{R}^{d_z}, \mathbf{W}_v \in \mathbb{R}^{d_m \times d_v}, \mathbf{b}_v \in \mathbb{R}^{d_v}$ are the learnable parameters, $\odot$ means the pointwise multiplication, and $d_v$ is the expanded hidden dimension which is set as $d_v = 2d_m$ in our work. Then the attention output is calculated as,

$$\mathbf{O} = f\left(\mathbf{Q}\mathbf{K}^T/s + \mathbf{B}_{\text{rel}}\right)\mathbf{V} \qquad \in \mathbb{R}^{r \times d_v} \tag{6}$$

where $f$ is the attention function, $s = r$ is the normalizing factor, $\mathbf{B}_{\text{rel}} \in \mathbb{R}^{n \times n}$ is the relative positional bias which is set as the simple relative position bias [SUV18; MZK$^+$23] by default. We also have the choice for using RoPE [SLP$^+$21] relative position embedding, and in this case, we don't apply relative position bias but use RoPE in its original form to transform the $\mathbf{Q}$ and $\mathbf{K}$ sequences. Squared ReLU is applied [SML$^+$21] for the attention function on the image and the speech tasks and Softmax for the text related tasks. By default, the relative positions are calculated based on the token positions in the original sentence $\mathbf{H}$ instead of the compressed sequence $\mathbf{H}^c$. For working memory mechanism, we conduct local attention between $\mathbf{Q}, \mathbf{K}, \mathbf{V}$ with the sliding window size (or the working memory size) $w$, and we set the normalizing factor $s = w$. The final output of the GAU is computed with a gating mechanism,

$$\mathbf{G} = \text{SiLU}(\mathbf{H}^c\mathbf{W}_g + \mathbf{b}_g) \qquad \in \mathbb{R}^{r \times d_v} \tag{7}$$

$$\mathbf{Y}^c = (\mathbf{G} \odot \mathbf{O})\mathbf{W}_h + \mathbf{b}_h \qquad \in \mathbb{R}^{r \times d_m} \tag{8}$$

where $\mathbf{W}_g \in \mathbb{R}^{d_m \times d_v}, \mathbf{W}_h \in \mathbb{R}^{d_v \times d_m}, \mathbf{b}_g \in \mathbb{R}^{d_v}, \mathbf{b}_h \in \mathbb{R}^{d_m}$ are learnable parameters.

## A.3 Details of Multi-dimensional Damped EMA

As a special kind of SSM, the Multi-dimensional Damped EMA (MD-EMA) operator is first proposed in the MEGA [MZK$^+$23] paper, and we briefly introduce it here for the completeness of our work. In Equation (2), the kernel $\mathbf{K} \in \mathbb{R}^{n \times d_m}$ of MD-EMA is parameterized as following,

$$\mathbf{K} = \left(\sum_{i=1}^{h} \boldsymbol{\eta}_i(\boldsymbol{\alpha} \odot \boldsymbol{\beta})_i, \sum_{i=1}^{h} \boldsymbol{\eta}_i(\boldsymbol{\phi} \odot \boldsymbol{\alpha} \odot \boldsymbol{\beta})_i, \ldots, \sum_{i=1}^{h} \boldsymbol{\eta}_i(\boldsymbol{\phi}^n \odot \boldsymbol{\alpha} \odot \boldsymbol{\beta})_i\right) \tag{9}$$

where $\boldsymbol{\eta} \in \mathbb{R}^{h \times d_m}, \boldsymbol{\alpha} \in [0,1]^{h \times d_m}, \boldsymbol{\delta} \in [0,1]^{h \times d_m}, \boldsymbol{\beta} \in \mathbb{R}^{h \times d_m}, \boldsymbol{\phi} = 1 - \boldsymbol{\alpha} \odot \boldsymbol{\delta}$, are learnable parameters, $\odot$ means the element-wise multiplication and $h$ is a hyperparamter denoted as the EMA dimension. With the pre-computed kernel $\mathbf{K}$, the output of MD-EMA can be efficiently computed with FFTs in a time complexity of $O(n \log n)$. During sequential decoding stage at the time step $t$, the output of MD-EMA is calculated as

$$u_{t,ij} = \beta_{ij}s_{t,j}, \ \mathbf{s}_t = [s_{t,j}] \in \mathbb{R}^{d_m}$$

$$\mathbf{u}_t = [u_{t,ij}] \in \mathbb{R}^{h \times d_m}$$

$$\mathbf{z}_t = \boldsymbol{\alpha} \odot \mathbf{u}_t + (1 - \boldsymbol{\alpha} \odot \boldsymbol{\delta}) \odot \mathbf{z}_{t-1}$$

$$\text{MD-EMA}(\mathbf{s}_t) = \sum_{i=1}^{h} \boldsymbol{\eta}_i\mathbf{z}_{t,i} + \mathbf{D} \odot \mathbf{s}_t$$

where $\mathbf{z}_0 = \mathbf{0}$, $\mathbf{s}_t \in \mathbb{R}^{d_m}$ is the input representation at time $t$, and $\mathbf{D} \in \mathbb{R}^{d_m}$ are learnable parameters. We can see that the inference of MD-EMA has $O(1)$ complexity per time step that is independent of the sequence length $n$.

## A.4 Details of Latent Configurator with Gumbel Softmax

To produce this baseline, we apply Gumbel Softmax to the probability computation in the latent configurator described in Section 3.3. Specifically, the modified probability calculation is as follow,

$$\mathbf{p}'_i = \frac{e^{(\mathbf{H}\mathbf{w}_i + b_i + G_i)/\tau'}}{e^{(\mathbf{H}\mathbf{w}_0 + b_0 + G_i)/\tau'} + e^{(\mathbf{H}\mathbf{w}_1 + b_1 + G_i)/\tau'}} \quad \text{for } i \in \{0, 1\}$$

where $G_i \sim \text{Gumbel}(0, 1)$ is the Gumbel noise sampled from a standard Gumbel distribution and $\tau'$ follows the following temperature annealing schedule as in the previous work [BZMA20],

$$\tau' = \max(2 \times 0.999995^T, 0.5),$$

where $T$ is the number of training steps.

Table 4: Hyper-parameter Settings of our SeqBoat model for the LRA benchmark and the Speech Command (SC) dataset. DP is the dropout rate, BSZ is batch size, LR is learning rate, WD is weight decay, and Pre-N is Pre-normalization. BN, LN and SN refer to Batch Normalization, Layer Normalization and Scale Normalization.

| Task | Depth | $d_m$ | $\alpha$ | $d_z$ | $d_v$ | Attn-FN | Norm | Pre-N | BSZ | LR | DP | WD | Epochs |
|---|---|---|---|---|---|---|---|---|---|---|---|---|---|
| **ListOps** | 6 | 80 | 0.3 | 64 | 160 | softmax | LN | FALSE | 64 | 0.004 | 0.1 | 0.001 | 60 |
| **Text** | 4 | 128 | 0.3 | 64 | 256 | softmax | SN | FALSE | 50 | 0.004 | 0.1 | 0.01 | 50 |
| **Retrieval** | 6 | 128 | 0.3 | 64 | 256 | softmax | SN | FALSE | 64 | 0.003 | 0.1 | 0.04 | 40 |
| **Image** | 8 | 160 | 0.4 | 96 | 320 | relu$^2$ | BN | TRUE | 50 | 0.01 | 0 | 0.02 | 200 |
| **Pathfinder** | 6 | 128 | 1.0 | 64 | 256 | relu$^2$ | BN | TRUE | 128 | 0.01 | 0 | 0.01 | 200 |
| **Path-X** | 6 | 128 | 1.0 | 64 | 256 | relu$^2$ | BN | TRUE | 128 | 0.01 | 0 | 0.01 | 100 |
| **SC-Raw** | 8 | 60 | 1.0 | 30 | 120 | relu$^2$ | BN | TRUE | 20 | 0.01 | 0 | 0.01 | 200 |

## A.5 Hyper-parameter settings

For Long Range Arena (LRA) and Speech Command tasks, we use the AdamW [LH18] optimizer and the detailed settings together with our initial temperature scale $\alpha$ for the latent configurator are shown in Table 4. Following S5 [SWL23], we use NLI-style fusion of encoder features for the Retrieval task of LRA. For SeqBoat model, we use a working memory size $w = 512$ for Path-X and $w = 256$ for other tasks in LRA. For Speech Command, a working memory size of 256 is used. And for language modeling tasks, we use the RAdam [LJH$^+$19] optimizer and a working memory of size 1,024. We use RoPE relative position embeddings and the Softmax attention function for language modeling, and the detailed hyperparameters are shown in Table 5. For the speech command task and the Text task of LRA, the relative positions are calculated based on the positions in the compressed sequence.

## A.6 Details of Efficiency Measurements

Due to the dynamic sparsity of our model, we first measure the training time as the average per step wall time across the full training stage. The training speed is then calculated as the inverse of the training time. The memory consumption is measured as the average per step GPU memory allocation across the full training stage. To ensure fair comparisons, the relative training speedup between different models are calculated based on the training time measured on the same hardware using the same batch size settings. All the experiments are conducted on a mixed cluster with 8 NVIDIA V100 32GB GPUs and 2 NVIDIA A5000 24GB GPUs.

An alternative way of efficiency measurement is to calculate the total Floating Point Operations (FLOPs) used for training. However, the FLOPs calculation does not take account of the time-consuming memory copying of large matrices (which is exactly the speed bottleneck of SMA due to the *scatter* operation), and thus we choose to follow the previous works to measure the speedup based on the actual wall time for a stricter and fairer comparison.

Table 5: Hyper-parameters of our SeqBoat model for language modeling.

| | enwik8 |
|---|---|
| Sequence Length $\times$ Batch Size | $8192 \times 8$ |
| Optimizer | RAdam |
| Learning Rate | 0.005 |
| Initial Learning Rate | 0.002 |
| RAdam-$\beta$ | (0.9, 0.98) |
| Learning Rate Decay | linear |
| Weight Decay | 0.1 |
| Dropout | 0.15 |
| Attention Dropout | 0.0 |
| Gradient Clipping | 0.25 |
| Warmup steps | 24K |
| Total updates | 400K |
| Decoder Layers | 14 |
| Model Size ($d_m$) | 616 |
| Query and Key Sizes ($d_z$) | 128 |
| Value Size ($d_v$) | 1232 |
| EMA dimension ($h$) | 16 |
| Initial Temperature Scale ($\alpha$) | 1.0 |
| Working Memory Size ($w$) | 1024 |
| Total Parameters | 39M |

## B Proof of Theorem 1

*Proof.* Given any $\mathcal{L}' \subseteq \mathcal{L} = \text{span}\{f_1^l, \ldots, f_M^l\}$, we want to show that there exists a pair $(\mathbf{a}_t', \mathbf{c}_t')$ such that $\mathcal{L}_{\text{SMA}}(\mathbf{a}_t', \mathbf{c}_t') = \mathcal{L}'$. Since $\mathcal{L}' \subseteq \mathcal{L}$, every function in $\mathcal{L}'$ can be written as a linear combination

of the functions $\{f_1^l, \ldots, f_M^l\}$. Let's consider a function $g \in \mathcal{L}'$. Then, we can write

$$g = \sum_{i=1}^{M} \beta_i f_i^l$$

where $\beta_i \in \mathbb{R}$. Now, let's define $\mathbf{a}_t'$ and $\mathbf{c}_t'$ as follows:

- $\mathbf{a}_t' = (a_t'^i)$ where $a_t'^i = 1$ if $\beta_i \neq 0$ and $a_t'^i = 0$ otherwise.
- $\mathbf{c}_t' = (c_t'^i)$ where $c_t'^i = 1$ if $\beta_i \neq 0$ and $c_t'^i = 0$ otherwise.

Then, we have

$$\mathcal{L}_{\text{SMA}}(\mathbf{a}_t', \mathbf{c}_t') = \left\{ \sum_{i \in I} \zeta_i c_t'^i f_i^l \mid \forall \zeta_i \in \mathbb{R}, I = \{i \mid a_t'^i = 1, 1 \leq i \leq M\} \right\}$$

Since $a_t'^i = 1$ and $c_t'^i = 1$ if and only if $\beta_i \neq 0$, then, $I = \{i \mid \beta_i \neq 0, 1 \leq i \leq M\}$, we have

$$\mathcal{L}_{\text{SMA}}(\mathbf{a}_t', \mathbf{c}_t') = \left\{ \sum_{i, \beta_i \neq 0} \zeta_i f_i^l \mid \forall \zeta_i \in \mathbb{R} \right\} = \left\{ \sum_{i=1}^{M} \beta_i f_i^l \mid \forall \beta_i \in \mathbb{R} \right\} = \mathcal{L}'$$

This completes the proof. $\qquad\qquad\qquad\qquad\qquad\qquad\qquad\qquad\qquad\qquad\qquad\square$

## C   Theorem of Implicit Regularization for Latent Decision Making

We first give a brief introduction of the implicit regularization to provide some backgrounds. The implicit regularization of gradient decent [BD20] is a side effect caused by the discreteness of the gradient descent updates. Specifically, it has been proved that a discrete gradient update $\theta' = \theta - h\nabla_\theta L(\theta)$ of the parameter $\theta$ is implicitly minimizing a transformed loss, $L' = L + \frac{h}{4}||\nabla_\theta L(\theta)||^2$ where $h$ is the learning rate and $L$ is the training loss term. Assuming the usage of Stochastic Gradient Decent (SGD) for gradient updates, [DMRB22] have shown that the implicit gradient regularizer $||\nabla_\theta L(\theta)||^2$ will conditionally put an implicit pressure on $||\nabla_\theta f_\theta(x)||_F^2$ for a multi-class classification cross-entropy loss term, where $f_\theta(x)$ is a neural network that takes the input $x$ and outputs the logits for classification. A Transfer Theorem [DMRB22] is further derived to characterize the conditional transfer of the implicit regularization pressure from $||\nabla_\theta f_\theta(x)||_F^2$ to the gradient norm $||\nabla_x f_\theta(x)||_F^2$, where $|| \cdot ||_F$ means the Frobenius norm. The transfer theorem assumes an ordinary neural architecture $f_\theta(x) : \mathbb{R}^d \mapsto \mathbb{R}^k$ with $l$ layers in the following form,

$$h_i(x) = a_i(w_{h_i} h_{i-1}(x) + b_i), \quad \text{for } i = 1, 2, \ldots, l,$$

where $a_i$ is the activation function, $h_i(x)$ indicates the sub-network from the input $x \in \mathbb{R}^d$ up to the output of layer $i$ and $h_0(x) = x$. This implies $h_l(x) = f_\theta(x)$.

In this work, we consider a neural architecture that has a per-layer latent configurator $p_i : \mathbb{R}^{|h_{i-1}|} \mapsto [0, 1], p_i(z) = y_i(w_i z)$, that maps the hidden representation from $h_{i-1}$ to a confidence probability of activating a submodule $q_i : \mathbb{R}^{|h_{i-1}|} \mapsto \mathbb{R}^{|h_i|}$, *i.e.*,

$$h_i(x) = a_i(p_i(h_{i-1}(x))q_i(h_{i-1}(x))), \quad \text{for } i = 1, 2, \ldots, l,$$

where $y_i$ is the normalizing function to produce the probability. We use a similar technique of perturbation argument [DMRB22; SLK$^+$18] for proving that the implicit pressure will also transfer to the latent configurator $p_i$, which leads to the implicit regularization of latent activation decisions.

**Theorem 2** (Transfer Theorem for Latent Configurator). *Let $f_\theta : \mathbb{R}^d \to \mathbb{R}^k$ represent a deep neural network consisting of $l$ layers, i.e., $f_\theta(x) = f_l \circ f_{l-1} \circ \cdots \circ f_1(x)$ where $f_i(z) = a_i(p_i(z)q_i(z))$, $p_i : \mathbb{R}^{|z|} \mapsto [0, 1]$, $p_i(z) = y_i(w_i z)$ is a latent configurator, $q_i : \mathbb{R}^{|z|} \mapsto \mathbb{R}^{|h_i|}$ is a sub-module function, $a_i$ is the activation function, $y_i$ is the probability normalization function and $w_i$ is a weight matrix. For $i = 1, 2, \ldots, l$, we have*

$$||\nabla_{w_i} r_i(x)||_2^2 \frac{||d_i(x)||_2^2 ||\nabla_x f_\theta(x)||_2^2}{||d_i(x)\nabla_x r_i(x) + r_i(x)\nabla_x d_i(x)||_2^2} \leq ||\nabla_{w_i} f_\theta(x)||_2^2 \qquad (10)$$

*where $r_i(x) := p_i(h_{i-1}(x))$ and $d_i(x) := q_i(h_{i-1}(x))$, $h_i(x)$ indicates the sub-network from the input $x \in \mathbb{R}^d$ up to the output of layer $i$ with $h_0(x) = x$ and $||\cdot||_2$ denotes the L2 operator norm for matrix and L2 norm for vector.*

*Proof.* Following the above notations, we first rewrite the model function as

$$f_{w_i}(x) = g_i(r_i(x)d_i(x)),$$

where $g_i(z)$ is the remaining network of the full model $f_\theta(x)$ following the $i$-th layer. Considering a small perturbation $x + \delta x$ of the input $x$, there exists a corresponding perturbation $w_i + u(x)$ for the weight matrix in the latent configurator $p_i$ so that

$$f_{w_i}(x + \delta x) = f_{w_i+u(x)}(x). \tag{11}$$

For sufficiently small $\delta x$, we can identify $r_i(x+\delta x)d_i(x+\delta x)$ with the linear approximation around $x$, *i.e.*,

$$r_i(x + \delta x)d_i(x + \delta x) = r_i(x)d_i(x) + d_i(x)\nabla_x r_i(x)\delta x + r_i(x)\nabla_x d_i(x)\delta x$$

where $\nabla_x\, d_i : \mathbb{R}^d \mapsto \mathbb{R}^{|h_i|}$ and $\nabla_x\, r_i : \mathbb{R}^d \mapsto [0,1]$ are the total derivatives. We have

$$\begin{aligned}
f_{w_i}(x + \delta x) &= g_i(r_i(x + \delta x)d_i(x + \delta x)) \\
&= g_i(r_i(x)d_i(x) + d_i(x)\nabla_x r_i(x)\delta x + r_i(x)\nabla_x d_i(x)\delta x),
\end{aligned}$$

and similarly, for $r_i(x) = y_i(w_i h_{i-1}(x))$, and the perturbation $w_i \to w_i + u(\delta x)$, we can have the linear approximation $r'_i(x) = y_i((w_i + u(\delta x))h_{i-1}(x)) = r_i(x) + \nabla_{w_i} r_i(x)u(\delta x)$. This implies

$$f_{w_i+u(\delta x)}(x) = g_i(r_i(x)d_i(x) + \nabla_{w_i} r_i(x)u(\delta x)d_i(x)).$$

Therefore, substituting both sides of Equation (11) with the equations above, and we get

$$\nabla_{w_i} r_i(x)u(\delta x)d_i(x) = d_i(x)\nabla_x r_i(x)\delta x + r_i(x)\nabla_x d_i(x)\delta x.$$

Note the fact that $\nabla_{w_i} r_i(x)^T \nabla_{w_i} r_i(x) = ||\nabla_{w_i} r_i(x)||_2^2$. This implies

$$u(\delta x) = \frac{\nabla_{w_i} r_i(x)^T (d_i(x)\nabla_x r_i(x)\delta x + r_i(x)\nabla_x d_i(x)\delta x)d_i(x)^T}{||\nabla_{w_i} r_i(x)||_2^2 ||d_i(x)||_2^2}. \tag{12}$$

Defining $u(\delta x)$ as above and taking the derivative of both sides of Equation (11) with respect to $\delta x$ at $\delta x = 0$, we have

$$\nabla_x f_{w_i}(x) = \nabla_{w_i} f_\theta(x)\frac{\nabla_{w_i} r_i(x)^T (d_i(x)\nabla_x r_i(x) + r_i(x)\nabla_x d_i(x))d_i(x)^T}{||\nabla_{w_i} r_i(x)||_2^2 ||d_i(x)||_2^2}.$$

Applying the squared L2 operator norm $||\cdot||_2^2$, at both sides of the equation, we get

$$\begin{aligned}
||\nabla_x f_{w_i}(x)||_2^2 &= \left\| \nabla_{w_i} f_\theta(x)\frac{\nabla_{w_i} r_i(x)^T (d_i(x)\nabla_x r_i(x) + r_i(x)\nabla_x d_i(x))d_i(x)^T}{||\nabla_{w_i} r_i(x)||_2^2 ||d_i(x)||_2^2} \right\|_2^2 \\
&\leq ||\nabla_{w_i} f_\theta(x)||_2^2 \frac{||d_i(x)\nabla_x r_i(x) + r_i(x)\nabla_x d_i(x)||_2^2}{||\nabla_{w_i} r_i(x)||_2^2 ||d_i(x)||_2^2}.
\end{aligned}$$

Since $\nabla_x f_{w_i}(x) = \nabla_x f_\theta(x)$, this completes the proof. $\qquad\square$

**Remarks** Considering the architecture of the latent configurator in the SeqBoat layer, *i.e.*, the activation confidence probability is calculated as

$$p_i(z) = \frac{e^{z \cdot w_i^1/\tau_i}}{e^{z \cdot w_i^0/\tau_i} + e^{z \cdot w_i^1/\tau_i}},$$

then we can derive the squared L2 norms of its derivatives,

$$||\nabla_{w_i^1} p_i(z)||_2^2 = ||\nabla_{w_i^0} p_i(z)||_2^2 = (p_i(z)(1 - p_i(z))||z||_2/\tau_i)^2,$$

$$||\nabla_z p_i(z)||_2^2 = (p_i(z)(1 - p_i(z))||w_i^0 - w_i^1||_2/\tau_i)^2.$$

Substituting the above gradient values to Equation (10), we can see that

$$||\nabla_{w_i} r_i(x)||_2^2 \frac{||d_i(x)||_2^2 ||\nabla_x f_\theta(x)||_2^2}{||d_i(x)\nabla_x r_i(x) + r_i(x)\nabla_x d_i(x)||_2^2} \le ||\nabla_{w_i} f_\theta(x)||_2^2$$

$$\implies \frac{||h_{i-1}(x)||_2^2 ||d_i(x)||_2^2 ||\nabla_x f_\theta(x)||_2^2}{||d_i(x)||_2^2 ||\nabla_x h_{i-1}(x)||_2^2 ||w_i^0 - w_i^1||_2^2 + \frac{||\nabla_x d_i(x)||_2^2 \tau_i^2}{(1-p_i(h_{i-1}(x)))^2}} \le ||\nabla_{w_i} f_\theta(x)||_2^2$$

This implies that the implicit regularization pressure will tend to maximize the temperature $\tau_i$ to increase the exploration of the latent decisions, while at the same time maximize the probability of modular activation or de-activation through maximizing $||w_i^0 - w_i^1||_2^2$ and $p_i(h_{i-1}(x))$ for exploiting the decisions. Therefore, there will be an implicit balancing between the exploration and the exploitation of the latent decisions, and we conjecture that this kind of balancing is the cause of the dynamic sparse activation patterns we observed from the experiments.

# D    Additional Experiment Results

Table 6: Per-task speed up and memory allocation reduction on the LRA benchmark.

| Models (Input Length) | ListOps (2,048) | Text (4,096) | Retrieval (4,000) | Image (1,024) | Path (1,024) | Path-X (16,384) |
|---|---|---|---|---|---|---|
| *Training Step Speed Up ($\uparrow$)* | | | | | | |
| MEGA | 1.00× | 1.00× | 1.00× | 1.00× | 1.00× | 1.00× |
| MEGA-chunk | 2.07× | 2.42× | 2.66× | 1.39× | 1.46× | 1.36× |
| **SeqBoat-full** | 1.79× | 2.14× | 2.44× | 1.45× | 1.27× | 1.24× |
| **SeqBoat** | 2.26× | 3.61× | 2.50× | 1.33× | 1.31× | 1.30× |
| *Training Step Memory Allocation ($\downarrow$)* | | | | | | |
| MEGA | 1.00× | 1.00× | 1.00× | 1.00× | 1.00× | 1.00× |
| MEGA-chunk | 0.34× | 0.29× | 0.31× | 0.75× | 0.69× | 0.82× |
| **SeqBoat-full** | 0.45× | 0.23× | 0.27× | 0.74× | 0.68× | 0.59× |
| **SeqBoat** | 0.20× | 0.16× | 0.23× | 0.74× | 0.70× | 0.89× |
| *Evaluation Step Speed Up ($\uparrow$)* | | | | | | |
| MEGA | 1.00× | 1.00× | 1.00× | 1.00× | 1.00× | 1.00× |
| MEGA-chunk | 2.15× | 2.33× | 2.57× | 1.48× | 1.50× | 2.52× |
| **SeqBoat-full** | 1.70× | 2.47× | 2.59× | 1.52× | 1.30× | 1.95× |
| **SeqBoat** | 2.67× | 3.55× | 2.69× | 1.43× | 1.38× | 1.93× |

## D.1    Per-task Efficiency Measures for LRA Benchmark

As shown in Table 6, we compare the efficiency of our SeqBoat-full and SeqBoat models with the Mega and its chunk variants on all six tasks of the LRA benchmark. The evaluation speed is measured as the average inference throughput of the best performing model on the validation set. More details of the measurement are included in Appendix A.6. We can see that our SeqBoat model is consistently and substantially more efficient than Mega, and it also has comparable efficiency with Mega-chunk but significantly better average accuracy on the LRA benchmark.

# E    Additional Analysis

**How do different model design choices affect the performance?**    We do the ablation study of our SeqBoat-full model on the Image and ListOps tasks of the LRA benchmark, as in Table 7, to have a general understanding of the influence of the design of the latent configurator and the SSM. "Always Activated" means that we don't use SMA and always activate GAU for conducting second-order self-attention. We can see that although the validation accuracy is slightly improved on both Image and Listops, the training efficiency drops significantly. We also explore different parameterization of the convolution kernel for the linear SSM and apply S4D [GGGR22] to replace the original MD-EMA [MZK+23] kernel. The S4D variant of our model has much better training efficiency but the quality

Table 7: Ablation study of the SeqBoat-full model on Image and ListOps of the LRA benchmark. We report the validation accuracy on the Image and ListOps tasks, together with the training speed up for each of the tasks compared with MEGA.

| Model | Image | Speed (↑) | ListOps | Speed (↑) |
|---|---|---|---|---|
| SeqBoat-full | 90.16 | 1.45× | 60.95 | 1.79× |
| - Always Activated | 90.70 | 1.25× | 62.10 | 1.11× |
| - Gumbel Softmax | 89.94 | 1.29× | 59.75 | 1.76× |
| - SSM with S4D | 82.84 | 1.88× | 59.25 | 2.45× |

are significantly worse than SeqBoat-full with the default hyperparameter setting. It is possible that a well-tuned S4D kernel can have better performance than the MD-EMA for the use case of SMA but we leave a more comprehensive study as in future work. We also explore different differentiable approximation of latent decision making by replacing Tempered Softmax with Gumbel Softmax [JGP17]. However, we don't observe the performance improvements over Tempered Softmax for our setting. More details of the implementation of Gumbel Softmax based latent configurator is shown in Appendix A.4.

Table 8: Ablation study of the SeqBoat model on Image, ListOps and Pathfinder of the LRA benchmark. We report the validation accuracy on each of the tasks, together with the training speed up compared with MEGA.

| Model | Image | Speed (↑) | ListOps | Speed (↑) | Path. | Speed (↑) |
|---|---|---|---|---|---|---|
| SeqBoat | 90.36 | 1.33× | 59.05 | 2.26× | 96.34 | 1.31× |
| SeqBoat-FFN | 89.60 | 1.13× | 56.45 | 2.05× | 96.62 | 0.97× |
| SeqBoat-{GAU, FFN} | 89.74 | 1.20× | 59.10 | 1.97× | 96.10 | 1.06× |
| SeqBoat-Sigmoid | 87.34 | 1.77× | 58.20 | 2.52× | 91.35 | 1.67× |
| SeqBoat-MoE | 85.60 | 1.66× | 58.10 | 2.19× | 91.41 | 1.80× |
| SeqBoat-local | 90.12 | 1.46× | 59.00 | 1.78× | 96.08 | 1.54× |
| SeqBoat-chunk | 86.46 | 2.17× | 58.35 | 2.79× | 93.91 | 2.97× |
| SeqBoat-mem32 | 78.16 | 1.73× | 58.70 | 2.14× | 92.97 | 1.86× |
| SeqBoat-local32 | 75.58 | 1.67× | 53.90 | 2.19× | 82.05 | 1.80× |

We further present additional ablation studies on our SeqBoat model to justify the effectiveness of both the proposed working memory mechanism and the design choices of our architecture. We explain the meaning of the postfixes in the table as following:

- "-FFN": We add an additional Feed-Forward layer (with the same architecture as MEGA) after each SeqBoat layer.

- "-{GAU, FFN}": We add a FFN module in parallel with the GAU module to have $M = 2$ with two modules in the function space $\mathcal{F} = \{GAU, FFN\}$ for our SMA mechanism to control the activation.

- "-Sigmoid": We use Sigmoid instead of Tempered Softmax for latent decision probability calculation.

- "-MoE": Instead of using our Tempered Softmax for module activation decisions, we adapt the X-MoE [JJNH91] routing mechanism to our use case by only considering two modules, a GAU module and a module that always outputs zero. We do not apply load balancing loss in this case because there is only one module that needs to do actual computation.

- "-local": We don't use SMA and always activate the GAU module with local attention.

- "-chunk": We don't use SMA and always activate the GAU module to process the equal-sized chunks of the input sequence. The chunk size is set to be the same as the memory size of SeqBoat for fair comparison.

- "-mem32": We restrict the memory size of SeqBoat to 32.

- "-local32": We restrict the sliding window size of SeqBoat-local to 32.

We first explore the application of SMA with multiple heterogeneous modules and identify the current engineering challenges. We can see that SeqBoat-{GAU, FFN} obtains better trade-offs than

SeqBoat-FFN (which simply appends an extra FFN layer after each SeqBoat layer) on both the Image and the ListOps tasks. However, SeqBoat-$\{\text{GAU}, \text{FFN}\}$ still falls behind SeqBoat (which does not include FFN anywhere at all). This is because the current implementation of SMA is I/O bounded due to copying large tensors with the *scatter* operator, and for a fair comparison with MEGA, we do not include any fused kernel to optimize the memory bandwidth utilization. We acknowledge that it needs more engineering efforts to scale up SMA with multiple modules, and advocate future works for the empirical impacts of SMA on the MoE community. We also include additional ablation results with X-MoE routing as an alternative design choice for the latent configurator. We can see that SeqBoat-MoE performs much worse than our SeqBoat model which uses the Tempered Softmax for decision confidence calculation.

Comparing SeqBoat-Sigmoid with SeqBoat, we can see that while theoretically using Softmax is equivalent to using Sigmoid under the binary decision scenario, using Softmax for decision confidence calculation gives much better results under our hyper-parameter settings. Comparing SeqBoat with SeqBoat-local and SeqBoat-chunk, we can see that our model obtains better accuracy than the models without SMA. If we constrain the memory size to a smaller value (see SeqBoat-mem32 v.s. SeqBoat-local32), this performance gap becomes more substantial (more than 10 percent absolute difference on Pathfinder). These evidences demonstrate the effectiveness of our SMA mechanism for capturing long-term dependencies. Comparing SeqBoat-FFN with SeqBoat, we can see that adding extra FFN layers does not provide any empirical performance gains for SeqBoat, and this motivates us to not include FFNs anywhere in our architecture.

# F  Related Work

**Dynamic Routing**  [RSVG20b] aims to alleviate the second-order complexity of self-attention by dynamically mapping each of the token to its nearest-neighbor clusters, and only do self-attention inside the clusters. Dynamic routing can be understood as the following form when it is applied for sequence modeling,

$$\max_{\theta, \mathbf{x}_{<t}^c} P_f(\mathbf{x}) = \prod_{t=1}^{n} P_f(x_t | \mathbf{x}_{<t}^c, \theta)$$

where $f$ is the Transformer architecture with self-attention layers, and it can be seen as a special case of our formulation described in Equation (1). Even though dynamic routing learns both the parameters and the corresponding context for each of the token prediction, it has a fixed architecture $f$ that is non-adaptive for both training and the inference.

**Mixture of Experts (MoE)**  [JJNH91; FZS22; LLX$^+$20; ZLL$^+$22] is designed to learn to select different parameters with the same architecture for each of the input elements. Specifically, it has the following form when applied to sequence modeling,

$$\max_{\theta_t} P_f(\mathbf{x}) = \prod_{t=1}^{n} P_f(x_t | \mathbf{x}_{<t}, \theta_t)$$

where the model parameter $\theta_t$ is selected from a pool of candidates at each time step $t$ based on the input content. We can see that it is a special case of our formulation as in Section 3.1 because it has a fixed context $\mathbf{x}_{<t}$ for each of the token prediction and the model architecture is non-adaptive.

**Comparison between SMA and MoE:**  The motivation behind our Sparse Modular Activation (SMA) mechanism is to enable neural networks to contextually skip modules of any architectures for efficient sequence modeling, while MoE aims to efficiently scale up the models with more parameters (usually by adding homogeneous modules). This difference of motivation results in a fundamental mechanism difference:

- SMA leverages a latent configurator to decide if each module needs to be activated for each sequence element, while MoE is designed to choose a predefined number of modules from a large group of modules.

This difference on the core design of the mechanisms further leads to the different properties of the mechanisms:

- SMA supports a dynamic model architecture that can learn to drop a sub-module entirely (for better efficiency) based on the task it is trained on, while MoE only selects the parameters of the modules for the same architecture. As shown in Figure 3, when the SeqBoat-full model is trained on the Text task of LRA, the first three layers learn to have zero activation time of the GAU module. This means that these layers degenerate to pure SSM layers from the initial SSM+GAU architecture after adapting to the target task.

- SMA is guaranteed to have a full coverage of the combinatorial search space ranging from zero activation to full activation of modules, while MoE can only cover a subset of the space by choosing a fixed number of modules to activate.

To the best of our knowledge, SMA is the first mechanism that successfully enables a neural network to obtain practical efficiency and complexity gains from sparsely activating a self-attention-like module, while none of the previous works on MoE ever achieved this.

**Comparison with Flash Attention [DFE$^+$22]:**   Flash Attention (FA) performs exact self-attention efficiently through utilizing hardware-aware techniques. FA and SMA focus on improving model efficiency on orthogonal levels. SMA is a module-level activation strategy that is perpendicular to how the attention module is actually doing the computation. In fact, we can also apply FA to the GAU module in our SeqBoat architecture for more efficient self-attention computation, but we decide not to use any custom fused kernels in our implementation for a fair comparison with MEGA and its variant.

**Comparison with Efficient Attention Strategies [KKL20; CLD$^+$20; WLK$^+$20; MKW$^+$21]:** Our SMA mechanism enables a dynamic architecture that can learn to drop a sub-module entirely for better efficiency when it is adapted to the target task, and we empirically observe such behavior for different tasks in LRA. This capability is not possible for the previous efficient attention strategies because their architectures are static and thus an efficient attention layer is always applied before the Feed-Forward layer. Also, SMA is an orthogonal mechanism to efficient attention approaches because it does not modify the attention itself but the input to the attention modules, and thus can be stacked with other efficient attention strategies.

**Comparison with GShard [LLX$^+$20]:**   GShard divides tokens into groups of the same size, while our SMA allows different modules to have different group size. This means SMA can support an adaptive neural architecture whose sub-modules can be completely dropped if no tokens are selected for that module. Also, while GShard is a routing mechanism, SMA is an activation mechanism that can still function when there is only one module under consideration.

**Comparison with Perceiver [JGB$^+$21]:**   Perceiver utilizes a pre-defined number of latent representations to repeatedly attend to the input array, which can be understood as conducting soft selection from the input array to a fixed subset. Our SMA operates on a different level to directly conduct a hard selection from the input to form a dynamic subset. Plus, our SMA can be applied to causal language modeling, while it is unclear how Perceiver can be adapted to this important use case in the era of large language model.

**Connection with Recurrent Independent Mechanisms [GLH$^+$19]:**   Generally, Recurrent Independent Mechanisms (RIMs) and our work both touch on how to learn sparsely interacting modules. However, our works focus more on the efficiency gains from the modular sparsity, while RIMs emphasize on the generalization benefits. Plus, while the GAU module in our work can be regarded as modeling the sparse interaction between the state representations of SSM at different time steps, our work does not explicitly try to use attention bottlenecks for modeling the interaction between different SSM modules. Instead, we apply a rather simple strategy of using linear aggregation to combine the representations from both the SSM and the GAU modules. It would be interesting to see if the attention bottleneck can help SMA to better scale up with multiple heterogeneous modules, and we leave it as a future work worth exploring.

# G  Limitations, Future Works & Broader Impact

Our study has some limitations that should be acknowledged. First, while our Sparse Modular Activation (SMA) mechanism is generally applicable for activating any sub-modules in neural networks, we have narrowed the scope of our paper to focus solely on developing a more efficient neural architecture with SSM and GAU for long sequence modeling, due to limited resources. Additionally, the current implementation of SMA is I/O bounded due to copying large tensors with the *scatter* operator. For a fair comparison with the previous models, we did not include any fused kernel to optimize memory bandwidth utilization. This limitation highlights the need for more engineering efforts to scale up SMA with multiple modules.

There are several promising directions for future research building on our work. One interesting avenue is to explore how SMA can boost the downstream performance of a pre-trained large model, given that pre-training a useful large language model from scratch can be prohibitively expensive. Additionally, since SMA can produce discrete and dynamic module activation patterns for each of the data sample, our work opens up new possibilities for interpretability in neural networks. Future research could investigate the correlations between the activation of specific modules and input tokens or model predictions, to gain a better understanding of the properties of tasks or the behavior of models. Furthermore, our mechanism may provide extra controllability over model predictions, which can be achieved by manually modifying the module activation during inference time. Finally, a promising future direction is to investigate how to scale up SMA with a large number of modules, potentially leading to empirical impacts of SMA on the research field of Mixture-of-Experts. Some initial ideas include setting an upper bound for the activation time of each module, or limiting the number of modules that can be activated at the same time.

The SMA and the SeqBoat architecture can bring about considerable societal impact by enhancing the efficiency and interpretability of sequence models. These advancements can result in better-performing automated systems, like open-domain chatbot and automatic speech recognition, potentially improving communication, fostering inclusivity, and making such services more accessible to various societal sectors. The enhanced interpretability of models can generate trust in AI systems, especially in sensitive sectors like healthcare and finance. Transparent AI models can assist professionals in these fields to make better-informed decisions, potentially leading to improved societal outcomes.

However, these improvements come with a notable limitation: our research was unable to conduct experiments on models with a large number of parameters due to computational resource constraints. As such, while the proposed mechanisms show promising results in the tested scenarios, their scalability and applicability to larger, more complex models remain uncertain. This limitation may restrict the broader applicability of our findings, especially for large-scale problems or when fine-grained control over the model's behavior is needed. Furthermore, ethical considerations such as data privacy need to be seriously considered as our work progresses. As the efficiency of these models increases, there's an increased potential for processing large amounts of personal data, which necessitates robust privacy safeguards. Future work can also focus on addressing these limitations and ethical concerns, to ensure the advantages of these models can be responsibly realized.

