# OpenReview forum: "Sparse Modular Activation for Efficient Sequence Modeling"
_NeurIPS.cc/2023/Conference — NeurIPS 2023 poster_

### Official Review · Reviewer_oaMm · 2023-06-10

**Soundness:** 3 good
**Presentation:** 2 fair
**Contribution:** 3 good
**Rating:** 6
**Confidence:** 4

**Summary:**

The paper introduces Sailboat which builds upon MEGA. MEGA uses a combination of an Exponential Moving Average (EMA) block (which can be interpreted as a specific parameterization of the kernel from SSM models) and Gated Attention Units (GAU). MEGA explores both full attention and chunked attention. Sailboat instead takes a modular/MoE approach. Each layer in Sailboat linearly combines the outputs of multiple EMA + GAU modules. Moreover, the GAU layer in each module dynamically takes as input only a subset of the original input; thus reducing per module computation cost.  Sailboat-mem is also proposed as a variant of Sailboat that uses local attention with a sliding window in each module. Sailbot-mem performs better than MEGA-chunked in LRA while being more efficient.

**Strengths:**

1. The routing of a subset of inputs for attention per module is an interesting way to simplify the complexity introduced by soft module selection per layer. The method used in the paper is relatively straightforward and can be used as a general strategy for modular networks and modular Transformers.
2. The paper explores the space of models hybridizing SSMs and Transformers which is an interesting space to explore.
3. The empirical performance upshot is decent in some tasks and there is some efficiency gain over MEGA/MEGA-chunk

**Weaknesses:**

1. There are several technical unclarities - see questions
2. A few more ablations can be done - what if we use selective attention without modularity? what if we use just modules with just local attention (no softmax-based dynamic subset input selection)? Or just chunks (like MEGA)? --- particularly simpler alternatives of input subset selection seem to be missing.
3. In my interpretation the main technical contribution seems to be in the space of modular architectures. The fact that the authors use SSM+GAU as modules are just an implementation detail (although the empirical results of the implementation are noteworthy). In that sense, it could have been good to have more comparison/discussion/analysis with respect to other existing strategies involving MoEs or modules.
4. The value of the strategy in the space of efficient attention strategies is also a bit obscure. For example, what if we use BigBird/Performer etc. attention strategy to replace the attention in GAU in MEGA?
5. The empirical results are good on 2-3 tasks in LRA compared to MEGA-chunk, but otherwise Sailboat/Sailboat-Mem performs close to MEGA with limited efficiency gain.
6. It seems MEGA uses a feed forward MLP after the GAU-like block whereas Sailboat-mem. Uncontrolled variables like that can be potential confounders preventing clear comparison. Could be good to show some results in LRA with MEGA-chunk variant (with controlled parameters) with the same kind of block as Sailboat with the same activation and everything.

-------
**Post-Rebuttal score update:**

I increased the soundness score to 3 (from 2), but kept the overall score still to 5. Reasons below.

* Some of the issues are resolved in the rebuttal and additional experiments like 6, 2.

* Some of the important technical details are more clear mitigating my concern in 1., but hopefully they will be clarified better in the paper.

* I was initially not too convinced by the rebuttal about the lack of comparison against other efficient attention in MEGA-framework (point 5).  But now I think, the contribution of the adaptive input sparsification can be seen as a sort of orthogonal mechanism to efficient attention. It does not modify the attention itself but the input to the attention - and thus can be stacked with other efficient attention. So I think, under that perspective, point 5 is not as critical. Although perceiver is a relatively less orthogonal possible comparison. For this point and the above points, I am increasing the soundness score to 3.

* Given the clarification that $M=1$ (no. of pre-defined modules), I think on the flip side some of my points under the strengths section do not really apply anymore. A modular mechanism with just one module amounts to just lacking any modular mechanism. So I wouldn't say that the method makes much of a move in the space of modular deep learning or MoE. A theoretical framework is introduced but not tested empirically. Having more modules can also bring forth other engineering challenges and increase compute - which can result in a worse trade-off than Mega.

* Overall, the input compression mechanism is still interesting but my main concern at present is the presentation and the overall positioning of the paper. The paper emphasizes quite a bit on the modular setup but it seems that all those parts can be skipped over and only the layer input sparsification via tempered softmax can be kept to be relevant to the experiments. Given that it's hard to find (or missing) that M=1, the paper also feels misleading and confusing. This is mainly a presentation issue, but still seems substantial to me. Without this, the paper is $6$ for me. But for the last two points, I am still keeping the score as $5$ at the moment.

**Update 2**

I increased the score to $6$ based on further discussions and the authors' expressed willingness for increased clarification on the positioning.

**Questions:**

1. How are c$_i$ and a$_i$ computed for modules?
2. I did not understand the hyperparameter setup for the modules. For example, how many modules are used per layer (what is $M$?)? Wouldn't adding multiple modules per layer increase the parameters (according to your notation it seems your modules are layer specific as opposed to having a common set of modules for every layer to select/combine or is that an incorrect interpretation)? How do you control for parameter counts then?
3. Can be good to contrast/compare (even if not empirically), in the paper, the difference between GShard [4] and this approach. As far as I understand, GShard also creates local token groups for the parallel processing of modules.
4. MEGA uses Feed forward after the GAU block. Is that removed in Sailboat? Any specific reason?


Missing references:

[5] uses Gumbel sigmoid as a selector mechanism to dynamically discard input before attention although not in a modular context and without exact discrete selection during training.


Minor:
* Are ζ$_i$ in the eqn. near L120 learnable parameters?
* Is \forall ∀ in the eqn. near L120 a typo?
* Is there any reason why the function/module weights (ζ$_i$c$_i$) are not normalized?
* Another extra comparison could be to use something like perciever [3] for creating some constant number of latent states per module from the SSM outputs.
* “second-order self-attention” -- second-order in which sense?
* Should be "instantiated" in L144
* I assume $a^l ∈ \mathbb{R}^n$ should be more specifically $a^l \in \\{0,1\\}^n$ in L168?
* Probably better to have an extended related works section in the appendix with more discussions. The current related work is a bit sparse.
* Note that there are pure SSM-based approaches (they are very recent works so I am not factoring this point into my decision) that show competitive performances too like BigS [1] and Hyena [2].  The cited paper that shows SSMs perform poorly on machine translation doesn't seem to use GLUs or other features that were found to be crucial for better NLP performance from SSMs.
* It could be good to clarify/discuss/contrast better on your paper the difference between your approach and "sparse attention" approaches


------

[1] Pretraining Without Attention, Wang et al. ArXiv 2023: https://arxiv.org/abs/2212.10544

[2] Hyena Hierarchy: Towards Larger Convolutional Language Models, Poli et al. ArXiv 2023: https://arxiv.org/abs/2302.10866

[3] Perceiver: General perception with iterative attention. Jaegle et al. ICML 2021

[4] GShard: Scaling Giant Models with Conditional Computation and Automatic Sharding, Lepikhin et al. ICLR 2021

[5] How Does Selective Mechanism Improve Self-Attention Networks?, Geng et al. ACL 2020

**Limitations:**

Limitations and social impacts are discussed in the appendix. It's more or less adequate.

---

> ### Author Rebuttal · Authors · 2023-08-09
>
> We would like to thank the reviewer for the positive review of our work and the constructive feedback on our manuscript. In the following, we address the remaining concerns to hopefully motivate a clear acceptance score.
>
> **Technical clarity:** Please refer to our answers to Q1, Q2 and Minors. We will also update the Related Work section to include extra comparisons with the sparse attention approaches and the missing reference.
>
> **More Ablations:** We add the new ablations in the Additional Ablation Study section in **Global Response**. Comparing Sailboat-mem with Sailboat-local and Sailboat-chunk, we can see that our model obtains better accuracy than the models without SMA. If we constrain the memory size to a smaller value (see Sailboat-mem32 v.s. Sailboat-local32), this performance gap becomes more substantial (more than 10 percent absolute difference on Pathfinder). These evidences demonstrates the effectiveness of our SMA mechanism.
>
> **Comparison with MoEs:** We add this comparison in the Comparison with Mixture of Experts section in **Global Response**. We also include the additional ablation results with X-MoE routing as an alternative design choice of latent configurator. We can see that X-MoE performs much worse than our Tempered Softmax in our use case.
>
> **Comparison with efficient attention strategies:**  As discussed in **Global Response**, our SMA mechanism enables a dynamic architecture that can learn to drop a sub-module entirely for better efficiency when it is adapted to the target task, and we empirically observe such behavior for different tasks in LRA. This capability is not possible for the previous efficient attention strategies because their architectures are static and thus an efficient attention layer is always applied before the feedforward layer.
>
> **Limited performance improvements on LRA compared with MEGA-chunk:** Please refer to the Updated Results for Long Range Arena section in **Global Response**. We do have substantial performance improvement over MEGA-chunk on 5 out of 6 tasks on LRA, and the average accuracy is 1.96 points higher. In Figure 5 of **Appendix D** in our paper, we also prove that our model can provide substantially better speed-quality trade-off than MEGA-chunk when varying different memory sizes.
>
> **Results of Sailboat-mem with feed forward MLP:** As shown in the Additional Ablation Study section in **Global Response**, we can see that both Sailboat-mem-ffn and Sailboat-chunk produce significantly worse performance than Sailboat-mem. The Sailboat-chunk model uses the same block as Sailboat but follow the MEGA-chunk to use the chunking strategy for sequence subset selection. Sailboat-mem-ffn follows the MEGA-chunk to add an additional FFN after each Sailboat layer.
>
> ---
>
> **Q1:** *How are $c_i$ and $a_i$ computed for modules?*
>
> **A1:** The latent configurator will produce $c_i$ and $a_i$ for each of the module. Please refer to L167-L172 for how the configurator is implemented in Sailboat.
>
> **Q2:** *I did not understand the hyperparameter setup for the modules ... ... How do you control for parameter counts then?*
>
> **A2:** As said in L109, $M$ is the number of predefined functions (or modules), and we set $M=1$ to include only one GAU module per layer, whose activation is controlled by the latent configurator. Yes, an interesting future direction is to investigate how to scale up SMA with a large number of modules. It is promising to try some primitive ideas such as setting an upper bound for the activation time of each module, or setting a maximum for the number of modules that can be activated at the same time. However, this direction is out of the scope of our paper, and we leave it as the future work.
>
> **Q3:**  *... ... difference between GShard [4] and this approach?*
>
> **A3:** GShard divides tokens into groups of the **same** size, while our SMA allows different modules to have different group size. This means SMA can support an adaptive neural architecture whose sub-modules can be completely dropped if no tokens are selected for that module. Also, while GShard is a routing mechanism, SMA is an activation mechanism that can still function when there is only one module under consideration.
>
> **Q4:** *Is FFN removed in Sailboat? Any specific reason?*
>
> **A4:** Yes, it is removed. This is because it is kind of redundant in its design as the GAU module and it doesn't provide any empirical performance gains for Sailboat according to our ablation study.
>
>
> **Minor:**
>
> - *Are $\zeta_i$ in the eqn. near L120 learnable parameters?*
>
>   As in L117, they are intended to represent a linear combination over the pre-defined functions. In the implementation of our Sailboat architecture, it is instantiated as the last linear layer in the GAU module.
>
> - *Is \forall $\forall$ in the eqn. near L120 a typo?*
>
>   No, it is not. $L’$ is the function space of a layer equipped with SMA and $M$ number of predefined functions. Thus, $\forall \zeta_i $ indicates the coefficients for a linear combination of the predefined functions.
>
> - *Is there any reason why the function/module weights ($\zeta_i c_i$) are not normalized?*
>
>   As in L121-123, we want to show that $L'$, the function space of a layer equipped with SMA, can recover the original function space $L$ defined in L109, if all the functions are activated. This won't be true if the module weights are normalized.
>
> - *... ... extra comparison could be to use something like perceiver [3] ... ...*
>
>   Nice catch! We will try this idea and see if it works out, but it seems out of the scope of this paper.
>
> - *“second-order self-attention” -- second-order in which sense?*
>
>   In the sense of the interactions between the input elements. Self-attention does pairwise comparison between tokens, which is a second-order interaction.
>
> - *Should be "instantiated" in L144*
>
>   Thanks! Fixed it.
>
> - *I assume $a^l \in \mathbb{R}^n$ should be more specifically $a^l \in \\{0, 1\\}^n$ in L168?*
>
>   Yes, you are right! Fixed it.

---

> > ### Comment · Reviewer_oaMm · 2023-08-10
> > **Part 1**
> >
> > > A2: As said in L109, $M$ is the number of predefined functions (or modules), and we set $M=1$
> >  to include only one GAU module per layer, whose activation is controlled by the latent configurator. Yes, an interesting future direction is to investigate how to scale up SMA with a large number of modules. It is promising to try some primitive ideas such as setting an upper bound for the activation time of each module, or setting a maximum for the number of modules that can be activated at the same time. However, this direction is out of the scope of our paper, and we leave it as the future work.
> >
> > This part has me a bit worried. Under M=1, it seems like the focus is not on the modular aspect and that most of Page 3 is not particularly relevant to what is experimentally demonstrated. The contents on page 3 seems mainly relevant and interesting if there are multiple modules to combine. (If I am understanding correctly in all your experiments you have only $f^l_1$ in Figure 1). The main experimental demonstrations, now, seem to boil down to testing a form of adaptive sparsification of input before attention (with the possibility to ignore all input - resulting in layer skipping).
> >
> > The theoretical framework for the utilization of multiple modules is still interesting but seems to mainly remain at the theoretical level. Normally, modular frameworks are relevant for introducing some choice protocol for selecting some sparse set of modules from a pre-defined set or larger modules or parallel utilization of multiple modules. Thereby, experimental demonstration of the effectiveness of modular frameworks mainly makes sense with M > 1. But this aspect is lost in practice here if the module set just has a single module.
> >
> > Overall, under this light, I think the paper could have been cleaner and stronger if it focused solely on the adaptive sparsification strategy through tempered softmax and compared it to other efficient transformers (eg. local attention, BigBird, Perciever, Linear Transformer etc.) within the EMA+GAU framework. Because right now, it's hard to disentangle how much it is ahead of other efficient transformers due to the incorporation of a MEGA-like framework and how much for the novel adaptive sparsification strategy.
> >
> > The contents from page 3 with generalization for multi-module mixing can be introduced in a separate paper and investigated empirically there.
> >
> > > More Ablations: We add the new ablations in the Additional Ablation Study section in Global Response. Comparing Sailboat-mem with Sailboat-local and Sailboat-chunk, we can see that our model obtains better accuracy than the models without SMA. If we constrain the memory size to a smaller value (see Sailboat-mem32 v.s. Sailboat-local32), this performance gap becomes more substantial (more than 10 percent absolute difference on Pathfinder). This evidence demonstrates the effectiveness of our SMA mechanism
> >
> > Thank you for the additional ablations. They are helpful and allay some of my concerns about the effect of FFN and others. The experiments against local attention show some evidence of the superiority of the proposed adaptive input sparsification compared to more basic efficient attention baselines.
> >
> > > Comparison with efficient attention strategies: As discussed in Global Response, our SMA mechanism enables a dynamic architecture that can learn to drop a sub-module entirely for better efficiency when it is adapted to the target task, and we empirically observe such behavior for different tasks in LRA. This capability is not possible for the previous efficient attention strategies because their architectures are static and thus an efficient attention layer is always applied before the feedforward layer.
> >
> > This is a good point. But:
> >
> > 1. There is still an empirical question about the accuracy/time trade-off. Yes, other efficient attention methods cannot fully drop the layer but how much speed gain is achieved because of that? Moreover, even if they do not drop layers - they still may or may not get higher accuracies within the SSM/EMA-GAU setup. The question still remains - how much we are gaining from adaptive sparsification as proposed?
> >
> > 2. While efficient attention by itself does not drop layers, there is a literature on layer skipping/early stopping - a few cited [1,2]. Thus, it is again a bit unclear where the proposed method stands against some prior layer-skipping strategy + efficient attention.
> >
> > [1] Reducing Transformer Depth on Demand with Structured Dropout - Fan et al. ICLR 2020
> >
> > [2] Accelerating Training of Transformer-Based Language Models with Progressive Layer Dropping - Zhang et al. NeurIPS 2020
> >
> > (Point 2. above is minor for me because I acknowledge considering all these variables is difficult in a single paper)
> >
> > > Limited performance improvements on LRA compared with MEGA-chunk
> >
> > > Results of Sailboat-mem with feed-forward MLP
> >
> > I acknowledge the responses to these points. I am not as much concerned with them anymore.

---

> > ### Comment · Reviewer_oaMm · 2023-08-10
> > **Part 2**
> >
> > >  Q1: How are $c_i$ and $a_i$ computed for modules?
> >
> > > A1: The latent configurator will produce $c_i$ and $a_i$ for each of the module. Please refer to L167-L172 for how the configurator is implemented in Sailboat.
> >
> > L167-L172 seems to explain how $a_l \in \\{0,1\\}^n$ ($n$ being the sequence size) is computed for input sparsification, but my question is how $a_t \in \\{0,1\\}^M$ (L114) is computed for *module sparsification*.
> >
> > Would it be correct to say $a^i_t = max(a^i_l)$ (where $i$ is the module id, $a^i_t \in \\{0,1\\}$ is the decision value for module $i$, and $a^i_l \in \\{0,1\\}^n$ is the decision values for the input tokens going into module $i$)?
> >
> > In other words, this would mean a module is not selected iff no input token is selected for that module (If no input token is selected $max(a^i_l)$ would be $0$, and if at least one input token is selected $max(a^i_l)$ would be $1$).
> >
> > If my interpretation is correct, I think this should be more explicitly and formally stated in the paper.
> >
> > -------
> >
> > Thank you for the other clarificatory points. I think A3 would be good to include in related works.

---

> ### Author Response · Authors · 2023-08-20
>
> Thanks for acknowledging both our mechanism as an orthogonal contribution to efficient attention, and the theoretical contribution of our framework to the field of MoE.
>
> **Regarding the positioning of our paper:** We want to emphasize that we provide a general and unified framework for multiple lines of works including adaptive input sparsification, adaptive computation time, dynamic routing, sparse attention and mixture of experts. We theoretically prove that under this framework, our SMA mechanism can provide a full coverage of the search space among the predefined modules. However, we acknowledge that it is infeasible to empirically validate the effectiveness of SMA in all these research fields and tasks in one paper, and, as indicated in the title of our paper, we narrow down our scope to only focus on applying SMA for efficient sequence modeling. We successfully demonstrated that, Sailboat, as a preliminary application of SMA (sparsely activating a GAU based on the state representation from SSM), can offer substantially better quality-efficiency trade-off than previous hybrid models. We also explore the application of SMA with multiple heterogeneous modules and identify the current engineering challenges, and advocate future works on scaling up SMA for empirical impacts on the MoE community.
>
> We will include this discussion into our paper for a clearer understanding of our contributions.
>
>
> **Regarding the concern of $M=1$:** We did try adding another FFN module in parallel with the GAU module to have $M=2$ with two modules in the function space $\mathcal{F} =\\{GAU, FFN\\}$ for our SMA mechanism to control the activations. The results are shown in the table below.
>
> | Method             | Image | Speed | ListOps | Speed | Path. | Speed |
> |--------------------|--------|------|--------|------|--------|------|
> | Sailboat-mem-{GAU,FFN} | 89.74 | 1.20 | 59.10 | 1.97 | 96.10 | 1.06 |
> | Sailboat-mem-ffn  | 89.60 | 1.13 | 56.45 | 2.05 | 96.62 | 0.97 |
> | Sailboat-mem  | 90.36 |  1.33×  |  59.05  | 2.26×  |  96.34 | 1.31×  |
> ||||||||
>
> We can see that Sailboat-mem-{GAU,FFN} obtains better trade-offs than Sailboat-mem-ffn (which simply appends an extra FFN layer after each Sailboat layer) on both the Image and the ListOps tasks. However, Sailboat-mem-{GAU,FFN} still falls behind Sailboat-mem (which does not include FFN anywhere at all). This is because the current implementation of SMA is I/O bounded due to copying large tensors with the scatter operator, and, for a fair comparison with MEGA, we do not include any fused kernel to optimize the memory bandwidth utilization. We acknowledge that it needs more engineering efforts to scale up SMA with multiple modules, and leave this as a future work.
>
> We will include this discussion into the ablation study section of our paper. We will also explicitly indicate that the effective number of modules $M=1$ is used for Sailboat in the method section for a clearer presentation.
>
>
> **Regarding the strength of our paper:** We want to emphasize that the strength of the contribution to modular networks mentioned in the review still holds. SMA is a straightforward modular activation method and can be used as a general strategy for modular networks. SMA also simplifies the complexity of the attention module by creating subsets of inputs, which is not explored by previous MoE works.
>
>
> **Comparison with Perceiver:** Perceiver utilizes a predefined number of latent representations to repeatedly attend to the input array, which can be understood as conducting soft selection from the input array to a fixed subset. Our SMA operates on a different level to directly conduct a hard selection from the input to form a dynamic subset. Plus, our SMA can be applied to causal language modeling, while it is unclear how Perceiver can be adapted to this important use case in the era of large language models.
>
> We will include this discussion into the related works section for a clearer connection with previous works.
>
>
> **Q5:** *Would it be correct to say $a_t^i =max(a_l^i)$ (where $i$ is the module id, $a_t^i \in \\{0, 1\\}$ is the decision value for module $i$, and $a_l^i \in \\{0, 1\\}^n$  is the decision values for the input tokens going into module $i$)?*
>
> **A5:** Yes, this interpretation is correct. If only one input token is selected for module $i$, the module $i$ will still be activated and to process that token only.
>
> Thanks for pointing out the valuable clarification questions. We have incorporated them into the method sections of our paper for clearer presentation.

---

> > ### Comment · Reviewer_oaMm · 2023-08-20
> > **Thank you**
> >
> > Thank you for the additional confirmation and clarification.
> >
> > I have decided to increase the score to $6$ assuming the authors update the paper as discussed.

---

### Official Review · Reviewer_8NeG · 2023-06-15

**Soundness:** 4 excellent
**Presentation:** 3 good
**Contribution:** 4 excellent
**Rating:** 6
**Confidence:** 4

**Summary:**

This paper introduces a framework for representation learning, which involves multiple modules that can be applied dynamically on different inputs. The authors implement this framework using a combination of linear state space models (SSMs) and a gated attention unit (GAU), also combining ideas from adaptive computation, and mixture of experts. Combined, the network, dubbed Sailboat, obtains a better speed-accuracy tradeoff compared to multiple strong baselines on a range of tasks.

==============================

Post rebuttal:


Thank you for your response and for the clarification.

I now have a better understanding of the main contributions, and have raised my score to 6. It would be great to include this discussion in the next version of the paper.

As to the specific responses, please note that results important enough to be included in the introduction and/or the conclusion should be included in the main text, not the appendix.

**Strengths:**

The technical and empirical contributions of this work more than justify acceptance to NeurIPS.
These include a very interesting and clever approach, combining multiple lines of research, and addressing many technical challenges, as well as an extensive and diverse set of experiments, showing the superiority of the proposed approach over previous approaches.

**Weaknesses:**

At the same time, the paper gives the impression of having been hastily written. There are some clarity issues regarding the proposed method (see below); the introduction mentions details that are never discussed in the paper, and there are some inconsistencies between the introductory paragraph of section 2 and the rest of the section. Combined, these raise some concern that other problems might have been overlooked.

Moreover, the human working memory is only loosely connected to the Sailboat-Mem idea. In particular, $w$ is within hundreds or thousands, which is far larger than 5-7, which is the human working memory, and thus completely unrelated.

Finally, there are unclear gaps between the superb results on LRA compared to the results in tables 3 & 4, which are comparable and not substantially different from the baselines (e.g., MEGA).

See more details below.

**Questions:**

+ I had a hard time understanding the main method:
* * The introduction mentions (#57-62) a method for efficient parallel implementation of the method using the scatter operation, but this is never discussed later.
- - The introductory paragraph of section 2 seems to describe something broader than what is currently below it in this section. E.g., section 2 does not touch on Sparse Modular Activation at all.
- - The F function in 3.1 is not entirely clear to me. First, if I understand correctly, the mapping assigns each word in the vocabulary a boolean value, which indicates whether it is used or not? Does this mean that the selection process works at the type level (i.e., each word is always selected or ignored, rather than this decision being dependent on the context)? Further, how is the vocabulary defined for non-textual tasks? Also, how does this function relate to the next part (e.g., $a_t$?)
- - In 3.2, what is $M$?
– I didn’t fully understand the Sailboat-Mem approach. Do the authors simply attend to the nearest $w$ tokens, and in this sense the idea is similar to standard hybrid approaches, or do they attend to these tokens and then apply SMA? Also, the comment in #207 (“... maintaining the ability of attending to the key tokens …”) wasn’t clear to me as well.

- The results in table 2 (LRA) show that the proposed method is dramatically faster and consumes far less memory than all baselines (including MEGA). However, in tables 3 & 4, the speed and memory values of the two approaches are comparable. Can you please explain the source of these differences? Is this because the context length is larger for these tasks? The speech experiments discuss a large context length of 16K, so that doesn’t seem like the reason.

- why do different models have different numbers of layers (#275)?

- In what sense is the proposed method able to improve interpretability? (conclusion, limitations section)

Typos and such:
- #52: sparsely map -> sparsely *maps*
- #86: is optimized is optimized
- #144: are instantiate -> are *instantiated*
- #236: LRA consists of five tasks: -> *six* tasks

---

> ### Author Rebuttal · Authors · 2023-08-09
>
> We would like to thank the reviewer for the positive review of our work and the constructive feedback on our manuscript. In the following, we address the remaining concerns to hopefully motivate a clear acceptance score.
>
> **Presentation clarity:** Please refer to our Answers to the Q1, Q2, Q3, Q4.
>
>
> **Loose connection between human working memory and the Sailboat-Mem idea:** Thanks for pointing this out. We will weaken the claim, “mimic the human working memory”, in the abstract, but we still want to emphasize the effectiveness of our working memory mechanism. In Figure 5 of the **Appendix D**, we do demonstrate that our working memory size can be as small as **16** but still maintain a reasonably good performance (>90%) on the Pathfinder task of LRA. In contrast, the traditional chunking based method simply fails the tasks with random guessing when the chunk size is less than 128. We use a larger working memory size for the final Sailboat-mem model simply because we want to achieve better performance under the settings of the LRA benchmark.
>
>
> **Unclear performance gaps between LRA and Speech Recognition/Language Modeling:** This is because we follow the notations of models in the original MEGA paper for the Table 3 & 4 of our paper, and the so-called MEGA models listed in the tables are actually MEGA-chunk variants. We have fixed this by explicitly denoting the MEGA models as MEGA-chunk for clearer understanding. Also, in the updated results table for Speech Recognition in **Global Response**, we do have much better performance and speed up than MEGA-chunk.
>
> ---
>
> **Q1:** *The introduction mentions (#57-62) a method for efficient parallel implementation of the method using the scatter operation, but this is never discussed later.*
>
> **A1:**  As in **Appendix A.1**, we do discuss the detailed parallel implementation using the scatter operator in the Pytorch-like code snippets, and such snippets are well documented with comments for how they work. The existence of such code snippets is also explicitly said as in Line 131-132 of our paper.
>
> **Q2:** *The introductory paragraph of section 2 seems to describe something broader than what is currently below it in this section. E.g., section 2 does not touch on Sparse Modular Activation at all.*
>
> **A2:** Section 2 provides the background and motivates us to propose a general formulation of Time-Variant Sequence Modeling in Section 3.1 that supports a dynamic neural architecture. Sparse Modular Activation (SMA) is then built upon the two assumptions proposed in Line 104-111 of Section 3.1, and we further show that, in Line 121-123, SMA can recover the original function space $L$ mentioned in Line 109 to justify the design of SMA.
>
> **Q3:** *The F function in 3.1 is not entirely clear to me... ... Also, how does this function relate to the next part (e.g., $a_t$?)*
>
> **A3:** Thanks for pointing this out. We acknowledge that $[0,1]^V$ in Line 105 should be corrected as $[0,1]^{n \times V}$.  $[0,1]^{n \times V}$ means a space of $n \times V$ dimensional matrices whose values are between 0 and 1, thus the $\mathcal{F}$ function means the whole chain-structured model that takes a sequence of tokens as input and outputs the probability distributions over the target vocabulary. This function introduces the intermediate function space $L$, which is then used for showing that our SMA can recover the original function space (as shown in Line 121).
>
> **Q4:** *In 3.2, what is $M$? – I didn’t fully understand the Sailboat-Mem approach. Do the authors simply attend to the nearest $w$ tokens, ... ... Also, the comment in #207 (“...maintaining the ability of attending to the key tokens …”) wasn’t clear to me as well.*
>
> **A4:** As show in Line 109, $M$ is the number of the pre-defined functions. As said in Line 201, we apply local attention on the **compressed sequence** which only contains the activated inputs for the GAU module. In this sense, we first apply SMA and then we attend to the nearest $w$ tokens in the compressed sequence. For Line 207, the unbounded attention span between the query and the key token is because two consecutive tokens in the compressed sequence can be from the positions that are far away from each other in the original sequence.
>
> **Q5:** *The results in table 2 (LRA) show that the proposed method is dramatically faster ... ... so that doesn’t seem like the reason.*
>
> **A5:**  Please refer to the **Unclear performance gaps between LRA and Speech Recognition/Language Modeling** paragraph above.
>
> **Q6:** *why do different models have different numbers of layers (#275)?*
>
> **A6:** This is because prior works (e.g. MEGA, S4) are using different numbers of layers for different tasks in LRA, we follow the setting of prior works to have a fair comparison.
>
> **Q7:** *In what sense is the proposed method able to improve interpretability? (conclusion, limitations section)*
>
> **A7:**  As shown in Figure 4 in **Appendix D**, our method can produce discrete and dynamic module activation patterns for each of the data sample. In this sense, our mechanism allows people to find the correlations between the activation of a specific module and the input tokens/model predictions, so that we can have better understanding of the property of the tasks or the behavior of the models. In fact, as shown in Figure 3, SMA allows us to answer the questions related to task properties such as "How much attention is needed for different sequence modeling tasks?" Also, since our latent decisions are discrete, our mechanism may also provide extra controllability over the model predictions. This can be achieved by doing manual modifications of the module activation during the inference time. Generally, our work opens new possibilities for interpretability which are worth exploring in the future work.
>
> **Typos and such:** Thanks for pointing out the typos. Fixed them.

---

### Official Review · Reviewer_kQJ9 · 2023-07-07

**Soundness:** 2 fair
**Presentation:** 2 fair
**Contribution:** 2 fair
**Rating:** 5
**Confidence:** 3

**Summary:**

This method introduces a novel architecture for long text modeling, which builds upon traditional linear state space models (SSMs). Since SSMs have shown inferior performance, combining SSMs with self-attention has become a popular approach. In this paper, several efficiency-related questions are considered in such settings, such as the amount of additional attention required for SSMs on a per-task basis and whether neural networks can dynamically activate their attention modules to achieve improved quality and efficiency trade-offs. These questions pose an interesting inquiry.

To address these questions, the authors propose a new architecture module that sparsely activates a Gated Attention Unit based on the state representations learned from an SSM. Through experiments on diverse tasks, the proposed approach demonstrates competitive results when compared to strong baselines.

**Strengths:**

The research question presented is intriguing. The hybrid combination of SSM and attention has gained popularity. However, the extent of additional attention required for SSMs on a per-task basis remains unclear. By reducing the reliance on attention, significant inference speedup can be achieved.

The selected baseline for comparison appears to be adequate. The authors conduct comprehensive experiments to demonstrate the improvements brought by their proposed approach. Additionally, the ablation study effectively showcases the effectiveness of specific components of the method.

**Weaknesses:**

The method section is difficult to comprehend. The notation should be clearly defined before its usage. Furthermore, the complex settings depicted in Figure 3 lack sufficient discussion on the motivation behind each function. It is unclear how these functions contribute to the final performance and the necessity of including them. If possible, it would be helpful to have a simplified implementation that still achieves competitive results.

More evidence is needed to support the performance of the proposed method. While the method shows similar effectiveness compared to other approaches, the claim of higher training speedup requires further explanation. The evaluation of speedup is not adequately discussed, and the observed speedup appears to be marginal when compared to strong baselines.

**Questions:**

Could you please provide more information on how the training speedup is computed?

In line 119, what does "c" represent? Additionally, in Figure 1, what does "h" denote?

It seems that Figure 2 is listed before Figure 1. The correct order should be Figure 1 followed by Figure 2.

What is the motivation behind the intricate setting of the latent configurator? Could you elaborate on how each function contributes to the final performance?

How does the method perform when applied to widely-used pre-trained settings?

---

> ### Author Rebuttal · Authors · 2023-08-09
>
> We would like to thank the reviewer for the constructive feedback on our manuscript. In the following, we address the remaining concerns to hopefully encourage a positive evaluation.
>
>
> **Presentation clarity of the method section:** Please refer to our answers to Question 2 and 3. We will provide more details in the next version and improve the presentation clarity.
>
> **Motivation of the architecture design:** The design priniciple behind our Sailboat model is to allow a faster sytem with weaker capability, (e.g. an SSM), to learn to activate a slower system with stronger capability (e.g. a GAU) on demand. We explain the design motivations of our latent configurator as following.
>
> For a sequence of length $n$, the design goal of the latent configurator is to generate the binary decision vector and the confidence probability vector, $a^l\in \mathbb{R}^n$ and $c^l \in \mathbb{R}^n$ respectively, for the GAU module so that we can sparsely activate the module with the decision vector while allowing the gradients to be backpropaged to the latent configurator through the confidence vector. The first Linear layer is applied to project the contextualized representation $\mathbf{H}^l\in \mathbb{R}^{n \times d_m}$ to a matrix whose last dimension is two.  This matrix is then normalized by a Tempered Softmax function on its last dimension to produce a probability matrix in $\mathbb{R}^{n \times 2}$ whose first column is the probability of not activating the module and the second column is the probability of activating. The decision vector $a^l$ and the confidence vector $c^l$ are then calculated by applying the argmax and the max operator respectively on the last dimension.
>
> From the above descriptions, we can see that **all the functions are necessary** to produce the final decision and the confidence vectors, so that the latent configurator can work as we expected. In this sense, it is not possible to ablate some functions in the configurator to see how they contribute to the final performance. However, we do try different alternative designs such as using Gumbel Softmax/X-MoE/Tempered Sigmoid instead of Tempered Softmax for probability matrix calculation, which are further explained in L300-304 and the additional ablation study in **Global Response**.
>
>
> **More evidence of model performance:** Please refer to the Updated Results section in **Global Response**. Generally, with updated results, our Sailboat-mem is now the **new state-of-the-art** among models with linear inference complexity, outperforming the pure SSM-based model, S5, with remarkably faster training speed and much less GPU memory consumption. We also add more detailed ablation studies to explain the speedup and performance.
>
>
> **Explanation of training speedup measurement:**  Since the dynamic module-level sparsity of our Sailboat model will affect the training speed throughout the training process, we first measure the training time as the average per step wall time across the full training stage. The training speed is then calculated as the inverse of the training time. To ensure fair comparisons, the relative training speedup between different models are calculated based on the training time measured on the same hardware using the same batch size settings. All the experiments are conducted on a mixed cluster with 8 NVIDIA V100 32GB GPUs and 2 NVIDIA A5000 24GB GPUs.
>
> ---
>
> **Question 1:** *Could you please provide more information on how the training speedup is computed?*
>
> **Answer:** Please refer to the **Explanation of training speedup measurement** paragraph above.
>
>
> **Question 2:** *In line 119, what does "c" represent? Additionally, in Figure 1, what does "h" denote?*
>
> **Answer:**  As explained in line 118, $\mathbf{c}_t$ is the confidence probability vector which stores the probability of decisions for all the modules at the time step $t$, so the symbol $\mathbf{c}$ represents a confidence matrix storing the probabilities for all the modules at all the time steps. In Figure 1, $\mathbf{h}\in \mathbb{R}^{d_m}$ is one of the vectors of the sequence representation matrix $\mathbf{H}\in \mathbb{R}^{n \times d_m}$.
>
> **Question 3:** *It seems that Figure 2 is listed before Figure 1. The correct order should be Figure 1 followed by Figure 2.*
>
> **Answer:** Thanks for pointing this out. We have fixed this by re-ordering the Figures.
>
>
> **Question 4:** *What is the motivation behind the intricate setting of the latent configurator? Could you elaborate on how each function contributes to the final performance?*
>
> **Answer:** Please refer to the **Motivation of the architecture design** paragraph above.
>
> **Question 5:** *How does the method perform when applied to widely-used pre-trained settings?*
>
> **Answer:**  While SMA is generally applicable for activating any sub-modules in neural networks, we narrow the scope of our paper (due to limited resources) to only focus on developing a more efficient neural architecture with SMA for long sequence modeling. It is interesting to see how SMA can boost the downstream performance of a pre-trained model, but we think it is out of the scope of this paper and leave it as a future work that is worth exploring. For the setting of Language Model (LM) pre-training, we do test our model on the competitive LM task, enwiki8, and obtain better results than the baseline MEGA model.

---

> > ### Comment · Reviewer_kQJ9 · 2023-08-22
> >
> > Thanks for the response. I will raise the score to 5.

---

### Official Review · Reviewer_KtV6 · 2023-07-08

**Soundness:** 4 excellent
**Presentation:** 3 good
**Contribution:** 3 good
**Rating:** 5
**Confidence:** 4

**Summary:**

The authors employ a hybrid model combining linear state space models and attention modules, while incorporating sparse attention within the attention modules to reduce memory usage and improve speed. The proposed method, Sailboat, outperforms previous approaches in terms of quality, speed, and memory efficiency. The sparse activation module in Sailboat utilizes temperature softmax to activate modules and aggregates output from each activated module weighted by softmax values. Additionally, the authors propose a variant called Sailboat-Mem, which performs window-sized attention on tokens assigned to each module, resulting in speed improvements and memory optimization, with a slight reduction in quality.

**Strengths:**

The paper demonstrates a clever utilization of a mixture of experts (MoEs) and routing techniques, resulting in notable speed improvements.

The proposed method achieves results comparable to state-of-the-art approaches while achieving increased speedups and reduced memory requirements.

The simplicity of the proposed method is a significant strength, as it only requires tuning a single temperature parameter in the softmax activation.

The ablation studies conducted in the research paper providing comparisons with other variants, such as gumbel-softmax and complete activation of modules, highlight the method's trade-off in terms of performance, speed, and memory utilization.


**Weaknesses:**

One notable weakness is the absence of a baseline comparison with Flash attention which performs exact attention but provide greater speedups while utilizing hardware aware techniques [https://arxiv.org/pdf/2205.14135.pdf].

Despite achieving speedups and reduced memory compared to MEGA, the proposed method does not demonstrate an improvement in performance over MEGA.

There are other variants of routing, such as Top-k and REINFORCE, which could potentially be compared with the proposed method. The absence of such comparisons limits the scope and comprehensiveness of the work. Please refer Top-k https://arxiv.org/abs/2101.03961,  REINFORCE https://arxiv.org/abs/2202.01169

**Questions:**

The equation introduced at line 120 lacks defined variables beforehand, making it challenging to interpret. Could you provide more clarity on this equation by defining all the variables involved?

The improvement of linear state-space models over transformers appears to be significant. Can you explain why this is the case? Are the number of parameters comparable between the two models?

Are the SSM kernels in the hybrid models identical? Maintaining consistency in the SSM kernels would offer clearer insights into the improvements achieved by sparse activation. Could you provide further information on this aspect?

Could you provide more details about the schedule used for the temperature softmax function in the work?

Can MEGA be considered as having only one attention module where all tokens are processed by it? In contrast, Sailboat divides tokens across a set of attention modules and performs attention only on those assigned to each module. As Sailboat's performance still lags behind MEGA, is there still potential for further improvements in routing? Can you provide the associated FLOPs (floating-point operations) for Sailboat and compare them with MEGA? Additionally, could you justify the memory reduction achieved by Sailboat? Is the reduction because the memory bottleneck is bs x num_tokens x num_tokens tensor?

In Section 3.2, binary decisions in SMA can be accomplished by computing p_i=0 using only w_0 and b_0, and p_i=1 can be deduced as 1-p_i=0. Could you please provide a justification for your implementation, particularly regarding the usage of w_0, b_0, w_1, and b_1, as it seems to deviate from conventional approaches?

There is a typo in Line 816 with the repeated phrase "is optimized." Could you please correct this error?

**Limitations:**

There are no explicit limitations beyond those discussed in the paper.

---

> ### Author Rebuttal · Authors · 2023-08-09
>
> We would like to thank the reviewer for the positive review of our work and the constructive feedback on our manuscript. In the following, we address the remaining concerns to hopefully motivate a clear acceptance score.
>
>
> **Comparison with Flash Attention(FA):** FA and SMA focus on improving model efficiency on orthogonal levels. SMA is a module-level activation strategy that is perpendicular to how the attention module is actually doing the computation. In fact, we can also apply FA to the GAU module in our Sailboat architecture for more efficient self-attention computation, but we decide **not** to use any custom fused kernels in our implementation for a fair comparison with MEGA and its variant.
>
> **Lack performance improvement over MEGA:** The MEGA model has quadratic inference complexity but our Sailboat only has sub-quadratic complexity. It is unfair to solely compare the performance, while ignoring the better scalability provided by Sailboat. The linear complexity variant of our model, Sailboat-mem, does provide significantly better average accuracy over MEGA-chunk on LRA with better speedup and memory consumption. In L598-L607 of **Appendix D**, we further demonstrate that Sailboat-mem can provide much better quality-speed trade-off than MEGA-chunk on the Image and Pathfinder tasks of LRA.
>
>
> **Comparison with other routing variants:**  We want to first point out that our method is **not** a routing mechanism, but instead an activation mechanism for neural modules. For more details on the comparisons with MoEs, please refer to **Global Response**. Moreover, although SMA and MoE are not directly comparable, we add an additional ablation study in **Global Response** by adapting the X-MoE routing mechanism to our use case, and show that it actually provides much worse performance than our SMA mechanism.
>
> ---
>
> **Question 1:** *The equation introduced at line 120 lacks defined variables beforehand, making it challenging to interpret. Could you provide more clarity on this equation by defining all the variables involved?*
>
> **Answer:**   $L'$ represents the function space of a layer equipped with SMA and $M$ number of predefined functions. According to Line 117, the function space $L'$ is a linear combination of the activated functions at layer $l$, thus $\zeta_i$ indicate the scalar of the linear combination. As said in Line 113-120, $a_t^i$ is the decision value for each sub-module, and $c_t^i$ is the confidence value for the decision, and $I$ means the set of the indices of the activated functions.
>
> **Question 2:** *The improvement of linear state-space models over transformers appears to be significant. Can you explain why this is the case? Are the number of parameters comparable between the two models?*
>
> **Answer:**  According to S4 paper, the number of parameters are tied between two models. While the exact reasons behind the performance improvement of linear state-space models over transformers are out of the scope of our paper, we here provide some related papers [1,2] for the reviewer to futher read on. Note that our work is perpendicular to the previous works on SSM, and as said in Line 165-167, we use SSM simply because it can provide an efficient computation of the recurrent states, plus it is interesting to investigate how much extra attention is needed beyond the SSM module.
>
> **Question 3:** *Are the SSM kernels in the hybrid models identical? Maintaining consistency in the SSM kernels would offer clearer insights into the improvements achieved by sparse activation. Could you provide further information on this aspect?*
>
> **Answer:**  Yes, we use the same MH-EMA kernel for all the layers as indicated in Line 164-165 of our paper.
>
> **Question 4:** *Could you provide more details about the schedule used for the temperature softmax function in the work?*
>
> **Answer:** As said in Line 170-172 of our paper, we didn’t apply any scheduling for the temperature parameter and set it as a learnable parameter with a specific initialization.
>
> **Question 5:** *Can MEGA be considered as having only one attention module where all tokens are processed by it? ... ... Is the reduction because the memory bottleneck is bs x num_tokens x num_tokens tensor?*
>
> **Answer:** Yes, MEGA can be considered as that. Yes, there are further potential for the improvements, since our design of the latent configurator is rather simple. Since the major speed bottleneck of SMA lies on copying a large matrix (which is not included in the FLOPs calculation) through the *scatter* operator, we follow the previous works to measure the speedup based on the actual wall time for a stricter and fairer comparison. The memory reduction achieved by Sailboat is because the GAUs are processing a shorter sequence, and for some layers of the Sailboat model, the GAUs are skipped entirely because they deactivated by the latent configurator all the time. This kind of skipping results in both a smaller bs x num_tokens x num_tokens tensor and fewer SiLU activation operations inside the GAUs.
>
> **Question 6:** *In Section 3.2, binary decisions in SMA can be accomplished by ... ... as it seems to deviate from conventional approaches?*
>
> **Answer:**  While theoretically using Softmax is equivalent to using Sigmoid under the binary decision case, we empirically find that using Softmax with both $w_0$ and $w_1$ gives much better results under our hyper-parameter settings. Please refer to the performance of the Sailboat-mem-sigmoid model under the Additional Ablation Study section in **Global Response** for more details.
>
> **Question 7:** *There is a typo in Line 816 ... ... this error?*
>
> **Answer:** Thanks for pointing this out.  We have fixed this typo by deleting the additional "is optimized".
>
> ---
>
> [1] Resurrecting recurrent neural networks for long sequences. Orvieto, Antonio, et al. arXiv 2023.
>
> [2] Simple hardware-efficient long convolutions for sequence modeling. Fu, Daniel Y., et al. arXiv 2023.

---

### Author Rebuttal · Authors · 2023-08-09

# Global Response

## Updated Results

For the Sailboat-mem model, we find that down-scaling the attention matrix $QK^T$ with the window size $w$ instead of the compressed sequence length $r$ can lead to substantially better results (as shown in Equation (5) near the Line 521 of **Appendix A.2**). Thus, we apply this change to Sailboat-mem for both the Long Range Arena and the Speech Command benchmark.

The table below shows the updated results for Long Range Arena (LRA), where * indicates a hybrid model. Our Sailboat-mem is now the new **state-of-the-art** among models with linear inference complexity, outperforming the pure SSM-based model, S5, with remarkably faster training speed and much less GPU memory consumption. Note that for a fair comparison with MEGA, we use the same MH-EMA based SSM for Sailboat-mem, and MH-EMA is generally considered as a simpler version of S4 and has worse modeling power than S5. The fact that we can still achieve performance improvements over S5 under this unfair setting of comparison demonstrates the effectiveness of our proposed SMA mechanism. We also include detailed efficiency measures for each task in the PDF.

| Model    | ListOps | Text  | Retr. | Image | Path. | Path-X | Avg. | Speed | Mem. |
|-|-|-|-|-|-|-|-|-|-|
| S4   | 59.10 | 86.53  | 90.94   | 88.48 | 94.01| 96.07 | 85.86 | 4.8×  | 0.14× |
| S4D-LegS | 60.47 | 86.18 | 89.46   | 88.19 | 93.06 | 91.95 | 84.89 | 6.1×  | 0.14×  |
| S5 | _62.15_ | 89.31 | **91.40**  | 88.00 | _95.33_  | **98.58**  | _87.46_ | 6.1× | 0.14×  |
| Liquid-S4 | **62.75**  | 89.02 | 91.20 | _89.50_ | 94.8  | 96.66  | 87.32   | 1.2×  | 0.17×  |
| H3*  | 57.50 | 88.20 | 91.00 | 87.30 | 93.00 | 91.80  | 84.80   | 6.0×  | 0.24×  |
| MEGA-chunk*  | 58.76  | **90.19** | 90.97 | 85.80   | 94.41 | 93.81 | 85.66 | 7.0×  | 0.09×  |
| **Sailboat-mem*** | 61.70  | _89.60_  | _91.28_ | **90.10**  | **96.35** | _96.68_ | **87.62** | **10.4×**   | **0.05×**   |
| | | | | | | | | | |

The table below shows the updated results for Speech Command.

| Model | #Param. | Acc. | Speed | Mem. |
|-|-|-|-|-|
| S4 | 300K | 97.50 |-|-|
| MEGA-chunk | 300K | 96.92 | 1.00× | 1.00× |
| **Sailboat-mem** | 293K | 97.35 | 1.32× | 0.44× |
||||||
## Additional Ablation Study

We present additional ablation studies on Sailboat-mem to justify the effectiveness of both the proposed working memory mechanism and the design choices of our architecture. We explain the meaning of the postfixes in the table as following:
 - “-ffn” means we add an additional feedforward layer (with the same architecture as MEGA) after each Sailboat layer
- “-sigmoid” means we use sigmoid instead of Tempered Softmax for latent decision probability calculation.
- “-moe” means that instead of using our Tempered Softmax for module activation decisions, we adapt the X-MoE [1] routing mechanism to our use case by only considering two modules, a GAU module and a module that always outputs zero. We do not apply load balancing loss in this case because there is only one module that needs to do actual computation.
 - “-local” means that we don't use SMA and always activate the GAU module with local attention.
- “-chunk” means that we don't use SMA and always activate the GAU module to process the equal-sized chunks of the input sequence.
- “-mem32” means we restrict the memory size of Sailboat-mem to 32.
- “-local32” means we restrict the sliding window size of Sailboat-local to 32.

| Model| Image| Speed| ListOps | Speed | Path.|Speed|
|-|-|-|-|-|-|-|
| Sailboat-mem| 90.36 | 1.33×| 59.05| 2.26× | 96.34 | 1.31× |
| Sailboat-mem-ffn|89.60| 1.13×| 56.45| 2.05× | 96.62 | 0.97× |
| Sailboat-mem-sigmoid | 87.34 |1.77×| 58.20 | 2.52× | 91.35 | 1.67× |
| Sailboat-mem-moe | 85.60 | 1.66× | 58.10 | 2.19× | 91.41| 1.80× |
| Sailboat-local | 90.12 | 1.46×| 59.00| 1.78× | 96.08 | 1.54× |
| Sailboat-chunk | 86.46 | 2.17× | 58.35| 2.79×| 93.91 | 2.97×|
| Sailboat-mem32 | 78.16 | 1.73×| 58.70| 2.14× | 92.97 | 1.86×|
| Sailboat-local32 | 75.58 | 1.67× | 53.90| 2.19× | 82.05 | 1.80× |
||||||||
## Comparison with Mixture of Experts (MoE)

The motivation behind our Sparse Modular Activation (SMA) mechanism is to enable neural networks to contextually skip modules of any architectures for efficient sequence modeling, while MoE aims to efficiently scale up the models with more parameters (usually by adding homogeneous modules). This difference of motivation results in a fundamental mechanism difference:
- SMA leverages a latent configurator to decide if each module needs to be **activated** for each sequence element, while MoE is designed to choose a predefined number of modules from a large group of modules.

This difference on the core design of the mechanisms further leads to the following consequences:
- SMA supports a dynamic model architecture that can learn to drop a sub-module entirely (for better efficiency) based on the task it is trained on, while MoE only selects the parameters of the modules for the same architecture. As shown in Figure 3 of our paper, when the Sailboat model is trained on the Text task of LRA, the first three layers learn to have **zero** activation time of the GAU module. This means that these layers degenerate to pure SSM layers from the initial SSM+GAU architecture after adapting to the target task.
- SMA is guaranteed to have a full coverage of the combinatorial search space ranging from zero activation to full activation of modules, while MoE can only cover a subset of the space by choosing a fixed number of modules to activate. This is further explained in L113-L123 of our paper.

To the best of our knowledge, SMA is the **first** mechanism that successfully enables a neural network to obtain practical efficiency and complexity gains from sparsely activating a self-attention-like module, while none of the previous works on MoE ever achieved this.

---

[1] On the Representation Collapse of Sparse Mixture of Experts. Chi et al. NeurIPS 2022.

---

### Decision · Program_Chairs · 2023-09-21

**Decision:**

Accept (poster)

**Comment:**

This paper presents Sparse Modular Activation (SMA), a general mechanism for sparsely and dynamically activating sub-modules in neural networks, addressing the challenges of combining attention with Linear State Space Models (SSMs). The authors introduce a novel neural architecture, Sailboat, which employs SMA to sparsely activate a Gated Attention Unit (GAU) based on state representations from an SSM. The proposed model demonstrates a significantly better quality-efficiency trade-off compared to existing hybrid models across various tasks. The reviewers are overall positive towards the contribution of this paper. I would recommend to accept this paper.